# Enhanced $\mathcal{H}$-Consistency Bounds

**Anqi Mao**                                                           AQMAO@CIMS.NYU.EDU
*Courant Institute of Mathematical Sciences, New York*

**Mehryar Mohri**                                                      MOHRI@GOOGLE.COM
*Google Research and Courant Institute of Mathematical Sciences, New York*

**Yutao Zhong**                                                        YUTAO@CIMS.NYU.EDU
*Courant Institute of Mathematical Sciences, New York*

**Editors:** Gautam Kamath and Po-Ling Loh

## Abstract

Recent research has introduced a key notion of $\mathcal{H}$-consistency bounds for surrogate losses. These bounds offer finite-sample guarantees, quantifying the relationship between the zero-one estimation error (or other target loss) and the surrogate loss estimation error for a specific hypothesis set. However, previous bounds were derived under the condition that a lower bound of the surrogate loss conditional regret is given as a convex function of the target conditional regret, without non-constant factors depending on the predictor or input instance. Can we derive finer and more favorable $\mathcal{H}$-consistency bounds? In this work, we relax this condition and present a general framework for establishing *enhanced $\mathcal{H}$-consistency bounds* based on more general inequalities relating conditional regrets. Our theorems not only subsume existing results as special cases but also enable the derivation of more favorable bounds in various scenarios. These include standard multi-class classification, binary and multi-class classification under Tsybakov noise conditions, and bipartite ranking.

**Keywords:** consistency, $\mathcal{H}$-consistency, surrogate loss, learning theory

## 1. Introduction

The design of accurate and reliable learning algorithms hinges on the choice of surrogate loss functions, since optimizing the true target loss is typically intractable. A key property of these surrogate losses is Bayes-consistency, which guarantees that minimizing the surrogate loss leads to the minimization of the true target loss in the limit. This property has been well-studied for convex margin-based losses in both binary (Zhang, 2004a; Bartlett et al., 2006) and multi-class classification settings (Tewari and Bartlett, 2007). However, this classical notion has significant limitations since it only holds asymptotically and for the impractical set of all measurable functions. Thus, it fails to provide guarantees for real-world scenarios where learning is restricted to specific hypothesis sets, such as linear models or neural networks. In fact, Bayes-consistency does not always translate into superior performance, as highlighted by Long and Servedio (2013).

Recent research has addressed these limitations by introducing $\mathcal{H}$-consistency bounds (Awasthi, Mao, Mohri, and Zhong, 2022a,b; Mao, Mohri, and Zhong, 2023f,b,e). These bounds offer non-asymptotic guarantees, quantifying the relationship between the zero-one estimation error (or other target loss) and the surrogate loss estimation error for a specific hypothesis set. While existing work has characterized the general behavior of these bounds (Mao et al., 2024a), particularly for smooth surrogates in binary and multi-class classification, their derivation has been restricted by certain assumptions. Specifically, previous bounds were derived under the condition that a lower bound of

the surrogate loss conditional regret is given as a convex function of the target conditional regret, without non-constant factors depending on the predictor or input instance. Can we derive finer and more favorable $\mathcal{H}$-consistency bounds?

In this work, we relax this condition and present a general framework for establishing *enhanced $\mathcal{H}$-consistency bounds* based on more general inequalities relating conditional regrets. Our theorems not only subsume existing results as special cases but also enable the derivation of tighter bounds in various scenarios. These include standard multi-class classification, binary and multi-class classification under Tsybakov noise conditions, and bipartite ranking.

The remainder of this paper is organized as follows. In Section 3, we prove general theorems serving as new fundamental tools for deriving enhanced $\mathcal{H}$-consistency bounds. These theorems allow for the presence of non-constant factors $\alpha$ and $\beta$ which can depend on both the hypothesis $h$ and the input instance $x$. They include as special cases previous $\mathcal{H}$-consistency theorems, where $\alpha \equiv 1$ and $\beta \equiv 1$. Furthermore, the bounds of these theorems are tight. In Section 4, we apply these tools to establish enhanced $\mathcal{H}$-consistency bounds for constrained losses in standard multi-class classification. These bounds are enhanced by incorporating a new hypothesis-dependent quantity, $\Lambda(h)$, not present in previous work. Next, in Section 5, we derive a series of new and substantially more favorable $\mathcal{H}$-consistency bounds under Tsybakov noise conditions. Our bounds in binary classification (Section 5.1) recover as special cases some past results and even improve upon some. Our bounds for multi-class classification (Section 5.2) are entirely new and do not admit any past counterpart even in special cases. To illustrate the applicability of our results, we instantiate them for common surrogate losses in both binary and multi-class classification.

In Section 6, we extend our new fundamental tools to the bipartite ranking setting (Section 6.1) and leverage them to derive novel $\mathcal{H}$-consistency bounds relating classification surrogate losses to bipartite ranking surrogate losses. We also identify a necessary condition for loss functions to admit such bounds. We present a remarkable direct upper bound on the estimation error of the RankBoost loss function, expressed in terms of the AdaBoost loss, with a multiplicative factor equal to the classification error of the predictor (Section 6.2). Additionally, we prove another surprising result with a different non-constant factor for logistic regression and its ranking counterpart (Section 6.3). Conversely, we establish negative results for such bounds in the case of the hinge loss (Section 6.4).

In Appendix F, we provide novel enhanced generalization bounds. We provide a detailed discussion of related work in Appendix A. We begin by establishing the necessary terminology and definitions.

## 2. Preliminaries

We consider the standard supervised learning setting. Consider $\mathcal{X}$ as the input space, $\mathcal{Y}$ as the label space, and $\mathcal{D}$ as a distribution over $\mathcal{X} \times \mathcal{Y}$. Given a sample $S = ((x_1, y_1), \ldots, (x_m, y_m))$ draw i.i.d. according to $\mathcal{D}$, our goal is to learn a hypothesis $h$ that maps $\mathcal{X}$ to a prediction space, denoted by pred. This hypothesis is chosen from a predefined hypothesis set $\mathcal{H}$, which is a subset of the family of all measurable functions, denoted by $\mathcal{H}_{\text{all}} = \{h \colon \mathcal{X} \to \text{pred} \mid h \text{ measurable}\}$. We denote by $\ell \colon \mathcal{H} \times \mathcal{X} \times \mathcal{Y} \to \mathbb{R}_+$ the loss function that measures the performance of a hypothesis $h$ on any pair $(x, y)$. Given a loss function $\ell$ and a hypothesis set $\mathcal{H}$, we denote by $\mathcal{E}_\ell(h) = \mathbb{E}_{(x,y) \sim \mathcal{D}}[\ell(h, x, y)]$ the generalization error and by $\mathcal{E}_\ell^*(\mathcal{H}) = \inf_{h \in \mathcal{H}} \mathcal{E}_\ell(h)$ the best-in-class generalization error. We further define the conditional error and the best-in-class condition error as $\mathcal{C}_\ell(h, x) = \mathbb{E}_{y|x}[\ell(h, x, y)]$ and $\mathcal{C}_\ell^*(\mathcal{H}, x) = \inf_{h \in \mathcal{H}} \mathcal{C}_\ell(h, x)$, respectively. Thus, the generalization error can be rewritten as

$\mathcal{E}_\ell(h) = \mathbb{E}_X[\mathcal{C}_\ell(h, x)]$. For convenience, we refer to $\mathcal{E}_\ell(h) - \mathcal{E}_\ell^*(\mathcal{H})$ as the *estimation error*, to $\mathcal{E}_\ell^*(\mathcal{H}) - \mathcal{E}_\ell^*(\mathcal{H}_{\mathrm{all}})$ as the *estimation error* and to $\Delta\mathcal{C}_{\ell,\mathcal{H}}(h, x) \coloneqq \mathcal{C}_\ell(h, x) - \mathcal{C}_\ell^*(\mathcal{H}, x)$ as the *conditional regret*.

Minimizing the target loss function, as specified by the learning task, is typically NP-hard. Instead, a surrogate loss function is often minimized. This paper investigates how minimizing surrogate losses can guarantee the minimization of the target loss function. We are especially interested in three applications: binary classification, multi-class classification, and bipartite ranking, although our general results are applicable to any supervised learning framework.

**Binary classification.** Here, the label space is $\mathcal{Y} = \{-1, +1\}$, and the prediction space is $\mathrm{pred} = \mathbb{R}$. The target loss function is the binary zero-one loss, defined by $\ell_{0-1}^{\mathrm{bi}}(h, x, y) = 1_{\mathrm{sign}(h(x)) \neq y}$, where $\mathrm{sign}(t) = 1$ if $t \geq 0$ and $-1$ otherwise. Let $\eta(x) = \mathbb{P}(Y = +1 \mid X = x)$ be the conditional probability of $Y = +1$ given $X = x$. The condition error can be expressed explicitly as $\mathcal{C}_\ell(h, x) = \eta(x)\ell(h, x, +1) + (1 - \eta(x))\ell(h, x, -1)$. Common surrogate loss functions include the margin-based loss functions $\ell_\Phi(h, x, y) = \Phi(yh(x))$, for some function $\Phi$ that is non-negative and non-increasing.

**Multi-class classification.** Here, the label space is $[n] \coloneqq \{1, \dots, n\}$, and the prediction space is $\mathrm{pred} = \mathbb{R}^n$ for some $n \in \mathbb{Z}_+$. Let $h(x, y)$ denote the $y$-th element of $h(x)$, where $y \in [n]$. The target loss function is the multi-class zero-one loss, defined by $\ell_{0-1}(h, x, y) = 1_{\mathsf{h}(x) \neq y}$, where $\mathsf{h}(x) = \mathrm{argmax}_{y \in \mathcal{Y}} h(x, y)$. An arbitrary but fixed deterministic strategy is used for breaking ties. For simplicity, we fix this strategy to select the label with the highest index under the natural ordering of labels. Let $p(y \mid x) = \mathbb{P}(Y = y \mid X = x)$ be the conditional probability of $Y = y$ given $X = x$. The condition error can be explicitly expressed as $\mathcal{C}_\ell(h, x) = \sum_{y \in \mathcal{Y}} p(y \mid x)\ell(h, x, y)$. Common surrogate loss functions include the max losses (Crammer and Singer, 2001), constrained losses (Lee et al., 2004), and comp-sum losses (Mao et al., 2023f).

**Bipartite ranking.** Here, the label space is $\mathcal{Y} = \{-1, +1\}$, and the prediction space is $\mathrm{pred} = \mathbb{R}$. Unlike the previous two settings, the goal here is to minimize the bipartite misranking loss $\mathsf{L}_{0-1}$, defined for any two pairs $(x, y)$ and $(x', y')$ drawn i.i.d. according to $\mathcal{D}$, and a hypothesis $h$: $\mathsf{L}_{0-1}(h, x, x', y, y') = 1_{(y-y')(h(x)-h(x'))<0} + \frac{1}{2}1_{(h(x)=h(x'))\wedge(y\neq y')}$. Let $\eta(x) = \mathbb{P}(Y = +1 \mid X = x)$ be the conditional probability of $Y = +1$ given $X = x$. Given a loss function $\mathsf{L}: \mathcal{H} \times \mathcal{X} \times \mathcal{X} \times \mathcal{Y} \times \mathcal{Y} \to \mathbb{R}_+$ and a hypothesis set $\mathcal{H}$, the generalization error and the condition error can be defined accordingly as $\mathcal{E}_\mathsf{L}(h) = \mathbb{E}_{(x,y)\sim\mathcal{D},(x',y')\sim\mathcal{D}}[\mathsf{L}(h, x, x', y, y')], \overline{\mathcal{C}}_\mathsf{L}(h, x, x') = \eta(x)(1-\eta(x'))\mathsf{L}(h, x, x', +1, -1) + \eta(x')(1 - \eta(x))\mathsf{L}(h, x, x', -1, +1)$. The best-in-class generalization error and best-in-class condition error can be expressed as $\mathcal{E}_\mathsf{L}^*(\mathcal{H}) = \inf_{h\in\mathcal{H}} \mathcal{E}_\mathsf{L}(h)$ and $\overline{\mathcal{C}}_\mathsf{L}^*(\mathcal{H}, x, x') = \inf_{h\in\mathcal{H}} \mathcal{C}_\mathsf{L}(h, x, x')$, respectively. The estimation error and conditional regret can be written as $\mathcal{E}_\mathsf{L}(h) - \mathcal{E}_\mathsf{L}^*(\mathcal{H})$ and $\Delta\overline{\mathcal{C}}_{\mathsf{L},\mathcal{H}}(h, x, x') = \overline{\mathcal{C}}_\mathsf{L}(h, x, x') - \overline{\mathcal{C}}_\mathsf{L}^*(\mathcal{H}, x, x')$, respectively. Common bipartite ranking surrogate loss functions typically take the following form: $\mathsf{L}_\Phi(h, x, x', y, y') = \Phi\left(\frac{(y-y')(h(x)-h(x'))}{2}\right)1_{y\neq y'}$, for some function $\Phi$ that is non-negative and non-increasing. Another choice is to use the margin-based loss $\ell_\Phi(h, x, y) = \Phi(yh(x))$ in binary classification as a surrogate loss. We will specifically be interested in the guarantees of minimizing $\ell_\Phi$ with respect to the minimization of $\mathsf{L}_\Phi$.

## 3. New fundamental tools for $\mathcal{H}$-consistency bounds

This section introduces new tools for deriving finer and more general $\mathcal{H}$-consistency bounds. We begin with a brief overview of $\mathcal{H}$-consistency.

**Background on $\mathcal{H}$-Consistency bounds.** A desirable property of surrogate loss functions is *Bayes-consistency* (Zhang, 2004a; Bartlett et al., 2006; Steinwart, 2007; Tewari and Bartlett, 2007). Bayes-consistency ensures that, asymptotically, minimizing a surrogate loss $\ell_1$ over all measurable functions, denoted by $\mathcal{H}_{\text{all}}$, leads to the minimization of the target loss function $\ell_2$ over the same function family:

$$\mathcal{E}_{\ell_1}(h_n) - \mathcal{E}_{\ell_1}^*(\mathcal{H}_{\text{all}}) \xrightarrow{n \to +\infty} 0 \implies \mathcal{E}_{\ell_2}(h_n) - \mathcal{E}_{\ell_2}^*(\mathcal{H}_{\text{all}}) \xrightarrow{n \to +\infty} 0.$$

However, Bayes-consistency is an asymptotic property, providing no guarantees for approximate minimizers. Additionally, it applies only to the family of all measurable functions, which is less relevant in practical scenarios where restricted hypothesis sets $\mathcal{H}$ are used. To address these limitations, Awasthi et al. (2022a,b) proposed a more refined framework, called $\mathcal{H}$-*consistency bounds*. These bounds provide upper bounds on the target estimation error in terms of the surrogate estimation error for a concave function $\Gamma \geq 0$ with $\Gamma(0) = 0$:

$$\mathcal{E}_{\ell_2}(h) - \mathcal{E}_{\ell_2}^*(\mathcal{H}) + \mathcal{M}_{\ell_2}(\mathcal{H}) \leq \Gamma\big(\mathcal{E}_{\ell_1}(h) - \mathcal{E}_{\ell_1}^*(\mathcal{H}) + \mathcal{M}_{\ell_1}(\mathcal{H})\big), \tag{1}$$

where $\mathcal{M}_\ell(\mathcal{H}) = \mathcal{E}_\ell^*(\mathcal{H}) - \mathbb{E}_X\big[\mathcal{C}_\ell^*(\mathcal{H}, x)\big] \geq 0$ represents the *minimizability gap*, which measures the difference between the best-in-class generalization error and the expected best-in-class conditional error. This concept can also be adapted to the bipartite ranking setting, with $\mathcal{M}_{\mathsf{L}}(\mathcal{H}) = \mathcal{E}_{\mathsf{L}}^*(\mathcal{H}) - \mathbb{E}_{(x,x')}\big[\overline{\mathcal{C}}_{\mathsf{L}}^*(\mathcal{H}, x, x')\big]$.

The minimizability gap is always upper bounded by the approximation error but it is generally a more fine-grained measure (Mao et al., 2023b). When $\mathcal{H} = \mathcal{H}_{\text{all}}$ or $\mathcal{E}_\ell^*(\mathcal{H}) = \mathcal{E}_\ell^*(\mathcal{H}_{\text{all}})$, the minimizability gaps vanish (Steinwart, 2007) leading to *excess error bounds* that imply Bayes-consistency, by taking the limit. However, in general, minimizability gaps are non-zero and represent an inherent quantity depending on the distribution and the hypothesis.

Thus, $\mathcal{H}$-consistency bounds provide a stronger and more informative guarantee than Bayes-consistency, since they are both non-asymptotic and specific to the hypothesis set $\mathcal{H}$ used. Note that, by the sub-additivity of a concave function $\Gamma \geq 0$, an $\mathcal{H}$-consistency bound also implies

$$\mathcal{E}_{\ell_2}(h) - \mathcal{E}_{\ell_2}^*(\mathcal{H}) \leq \Gamma\big(\mathcal{E}_{\ell_1}(h) - \mathcal{E}_{\ell_1}^*(\mathcal{H})\big) + \Gamma(\mathcal{M}_{\ell_1}(\mathcal{H})) - \mathcal{M}_{\ell_2}(\mathcal{H}),$$

where $\Gamma(\mathcal{M}_{\ell_1}(\mathcal{H})) - \mathcal{M}_{\ell_2}(\mathcal{H})$ is an inherent constant depending on the hypothesis set and distribution. The ultimate algorithmic goal when using a surrogate loss $\ell_1$ is to minimize the estimation loss $\big[\mathcal{E}_{\ell_1}(h) - \mathcal{E}_{\ell_1}^*(\mathcal{H})\big]$. An $\mathcal{H}$-consistency bound ensures that reducing this error to $\epsilon$ implies that the target estimation loss $\big[\mathcal{E}_{\ell_2}(h) - \mathcal{E}_{\ell_2}^*(\mathcal{H})\big]$ is upper bounded by $\Gamma(\epsilon) + \Gamma(\mathcal{M}_{\ell_1}(\mathcal{H})) - \mathcal{M}_{\ell_2}(\mathcal{H})$, or just $\Gamma(\epsilon)$ when the minimizability gaps vanish. Recent work by Mao et al. (2024a) shows that for all smooth surrogate losses in binary classification, $\Gamma(\epsilon)$ behaves as $\sqrt{\epsilon}$ near zero.

**Enhanced $\mathcal{H}$-consistency bounds and tools.** While $\mathcal{H}$-consistency bounds offer strong, non-asymptotic guarantees tailored to $\mathcal{H}$, they can be further enhanced by considering a more general form such as the following:

$$\mathcal{E}_{\ell_2}(h) - \mathcal{E}_{\ell_2}^*(\mathcal{H}) + \mathcal{M}_{\ell_2}(\mathcal{H}) \leq \Gamma\big(\gamma(h)\big(\mathcal{E}_{\ell_1}(h) - \mathcal{E}_{\ell_1}^*(\mathcal{H}) + \mathcal{M}_{\ell_1}(\mathcal{H})\big)\big), \tag{2}$$

where $\gamma(h)$ is a factor depending on the hypothesis $h$. This refinement allows the bound to incorporate $h$-dependent information, enabling the use of more favorable functions $\Gamma$, which can improve

the bound's behavior near zero. In the following sections, we will demonstrate this for both classification and bipartite ranking. For instance, we will show that under certain noise conditions in classification, the behavior of $\Gamma$ can outperform the typical square-root dependence, approaching near-linear behavior.

The foundation of earlier $\mathcal{H}$-consistency bounds involves finding a convex function $\Psi$ or a concave function $\Gamma$ such that: $\Psi\big(\Delta\mathcal{C}_{\ell_2,\mathcal{H}}(h,x)\big) \leq \Delta\mathcal{C}_{\ell_1,\mathcal{H}}(h,x)$ or $\Delta\mathcal{C}_{\ell_2,\mathcal{H}}(h,x) \leq \Gamma\big(\Delta\mathcal{C}_{\ell_1,\mathcal{H}}(h,x)\big)$. We extend this approach by relaxing the inequalities to incorporate functions $\alpha(h,x)$ and $\beta(h,x)$ that depend on both the hypothesis and the input instance. The following two theorems illustrate this enhancement with general guarantees of the form (2) derived from such relaxed inequalities, where $\gamma(h)$ is defined in terms of $\alpha$ and $\beta$.

**Theorem 1** *Assume that there exist a convex function $\Psi\colon \mathbb{R}_+ \to \mathbb{R}$ and two positive functions $\alpha\colon \mathcal{H} \times \mathcal{X} \to \mathbb{R}_+^*$ and $\beta\colon \mathcal{H} \times \mathcal{X} \to \mathbb{R}_+^*$ with $\sup_{x\in\mathcal{X}} \alpha(h,x) < +\infty$ and $\mathbb{E}_{x\in\mathcal{X}}[\beta(h,x)] = 1$ for all $h \in \mathcal{H}$ such that the following holds for all $h \in \mathcal{H}$ and $x \in \mathcal{X}$: $\Psi\bigg(\dfrac{\Delta\mathcal{C}_{\ell_2,\mathcal{H}}(h,x)}{\beta(h,x)}\bigg) \leq \alpha(h,x)\,\Delta\mathcal{C}_{\ell_1,\mathcal{H}}(h,x)$. Then, the following inequality holds for any hypothesis $h \in \mathcal{H}$:*

$$\Psi\big(\mathcal{E}_{\ell_2}(h) - \mathcal{E}_{\ell_2}^*(\mathcal{H}) + \mathcal{M}_{\ell_2}(\mathcal{H})\big) \leq \gamma(h)\big(\mathcal{E}_{\ell_1}(h) - \mathcal{E}_{\ell_1}^*(\mathcal{H}) + \mathcal{M}_{\ell_1}(\mathcal{H})\big). \tag{3}$$

*with $\gamma(h) = [\sup_{x\in\mathcal{X}} \alpha(h,x)\beta(h,x)]$. If, additionally, $\mathcal{X}$ is a subset of $\mathbb{R}^n$ and, for any $h \in \mathcal{H}$, $x \mapsto \Delta\mathcal{C}_{\ell_1,\mathcal{H}}(h,x)$ is non-decreasing and $x \mapsto \alpha(h,x)\beta(h,x)$ is non-increasing, or vice-versa, then, the inequality holds with $\gamma(h) = \mathbb{E}_X[\alpha(h,x)\beta(h,x)]$.*

**Theorem 2** *Assume that there exist a concave function $\Gamma\colon \mathbb{R}_+ \to \mathbb{R}$ and two positive functions $\alpha\colon \mathcal{H} \times \mathcal{X} \to \mathbb{R}_+^*$ and $\beta\colon \mathcal{H} \times \mathcal{X} \to \mathbb{R}_+^*$ with $\sup_{x\in\mathcal{X}} \alpha(h,x) < +\infty$ and $\mathbb{E}_{x\in\mathcal{X}}[\beta(h,x)] = 1$ for all $h \in \mathcal{H}$ such that the following holds for all $h \in \mathcal{H}$ and $x \in \mathcal{X}$: $\dfrac{\Delta\mathcal{C}_{\ell_2,\mathcal{H}}(h,x)}{\beta(h,x)} \leq \Gamma\big(\alpha(h,x)\,\Delta\mathcal{C}_{\ell_1,\mathcal{H}}(h,x)\big)$. Then, the following inequality holds for any hypothesis $h \in \mathcal{H}$:*

$$\mathcal{E}_{\ell_2}(h) - \mathcal{E}_{\ell_2}^*(\mathcal{H}) + \mathcal{M}_{\ell_2}(\mathcal{H}) \leq \Gamma\big(\gamma(h)\big(\mathcal{E}_{\ell_1}(h) - \mathcal{E}_{\ell_1}^*(\mathcal{H}) + \mathcal{M}_{\ell_1}(\mathcal{H})\big)\big), \tag{4}$$

*with $\gamma(h) = [\sup_{x\in\mathcal{X}} \alpha(h,x)\beta(h,x)]$. If, additionally, $\mathcal{X}$ is a subset of $\mathbb{R}^n$ and, for any $h \in \mathcal{H}$, $x \mapsto \Delta\mathcal{C}_{\ell_1,\mathcal{H}}(h,x)$ is non-decreasing and $x \mapsto \alpha(h,x)\beta(h,x)$ is non-increasing, or vice-versa, then, the inequality holds with $\gamma(h) = \mathbb{E}_X[\alpha(h,x)\beta(h,x)]$.*

We refer to Theorems 1 and 2 as *new fundamental tools* because they incorporate additional factors, $\alpha$ and $\beta$, which depend on both the hypothesis $h$ and the instance $x$. These theorems generalize previous results from (Awasthi et al., 2022a,b), which can be recovered as special cases when $\alpha \equiv 1$ and $\beta \equiv 1$. Compared to earlier approaches, these new tools offer more precise $\mathcal{H}$-consistency bounds in familiar settings and extend them to new scenarios where previous methods are insufficient. We will demonstrate their applications in both contexts. Moreover, the bounds derived using these tools are *tight*.

**Lemma 3** *The bounds of Theorems 1 and 2 are* tight *in the following sense: for some distributions, Inequality (3) (respectively Inequality (4)) is the tightest possible $\mathcal{H}$-consistency bound that can be derived under the assumption of Theorem 1 (respectively Theorem 2).*

Note that when $\Gamma(0) \geq 0$, the concave function $\Gamma$ is sub-additive over $\mathbb{R}_+$, and the theorem implies the following inequality:

$$\mathcal{E}_{\ell_2}(h) - \mathcal{E}_{\ell_2}^*(\mathcal{H}) + \mathcal{M}_{\ell_2}(\mathcal{H}) \leq \Gamma\big(\gamma(h)\big(\mathcal{E}_{\ell_1}(h) - \mathcal{E}_{\ell_1}^*(\mathcal{H})\big)\big) + \Gamma(\gamma(h)\,\mathcal{M}_{\ell_1}(\mathcal{H})),$$

The bound implies that if the surrogate estimation loss of a predictor $h$ is reduced to $\epsilon$, then the target estimation loss is bounded by $\Gamma(\gamma(h)\,\epsilon) + \Gamma(\gamma(h)\,\mathcal{M}_{\ell_1}(\mathcal{H})) - \mathcal{M}_{\ell_2}(\mathcal{H})$. When the minimizability gaps are zero, for example when the problem is realizable, the upper bound simplifies to $\Gamma(\gamma(h)\,\epsilon)$. In the special case of $\Psi(x) = x^s$ or equivalently, $\Gamma(x) = x^{\frac{1}{s}}$, for some $s \geq 1$ with conjugate number $t \geq 1$, that is $\frac{1}{s} + \frac{1}{t} = 1$, we can further obtain the following result.

**Theorem 4** *Assume that there exist two positive functions $\alpha \colon \mathcal{H} \times \mathcal{X} \to \mathbb{R}_+^*$ and $\beta \colon \mathcal{H} \times \mathcal{X} \to \mathbb{R}_+^*$ with $\sup_{x \in \mathcal{X}} \alpha(h, x) < +\infty$ and $\mathbb{E}_{x \in \mathcal{X}}[\beta(h, x)] = 1$ for all $h \in \mathcal{H}$ such that the following holds for all $h \in \mathcal{H}$ and $x \in \mathcal{X}$: $\frac{\Delta \mathcal{C}_{\ell_2, \mathcal{H}}(h, x)}{\beta(h, x)} \leq \big(\alpha(h, x)\,\Delta \mathcal{C}_{\ell_1, \mathcal{H}}(h, x)\big)^{\frac{1}{s}}$, for some $s \geq 1$ with conjugate number $t \geq 1$, that is $\frac{1}{s} + \frac{1}{t} = 1$. Then, for $\gamma(h) = \mathbb{E}_X\Big[\alpha^{\frac{t}{s}}(h, x)\beta^t(h, x)\Big]^{\frac{1}{t}}$, the following inequality holds for any $h \in \mathcal{H}$:*

$$\mathcal{E}_{\ell_2}(h) - \mathcal{E}_{\ell_2}^*(\mathcal{H}) + \mathcal{M}_{\ell_2}(\mathcal{H}) \leq \gamma(h)\big[\mathcal{E}_{\ell_1}(h) - \mathcal{E}_{\ell_1}^*(\mathcal{H}) + \mathcal{M}_{\ell_1}(\mathcal{H})\big]^{\frac{1}{s}}.$$

As above, by the sub-additivity of $x \mapsto x^{\frac{1}{s}}$ over $\mathbb{R}_+$, the bound implies

$$\mathcal{E}_{\ell_2}(h) - \mathcal{E}_{\ell_2}^*(\mathcal{H}) + \mathcal{M}_{\ell_2}(\mathcal{H}) \leq \gamma(h)\Big[\big(\mathcal{E}_{\ell_1}(h) - \mathcal{E}_{\ell_1}^*(\mathcal{H})\big)^{\frac{1}{s}} + \big(\mathcal{M}_{\ell_1}(\mathcal{H})\big)^{\frac{1}{s}}\Big].$$

The proofs of Theorems 1, 2, 4, and Lemma 3 are presented in Appendix B. These proofs are more complex than their counterparts for earlier results in the literature due to the presence of the functions $\alpha$ and $\beta$. Our proof technique involves a refined application of Jensen's inequality tailored to the $\beta$ function, the use of Hölder's inequality adapted for the $\alpha$ function, and the application of the FKG Inequality in the second part of both Theorems 1 and 2. The proof of Theorem 4 also leverages Hölder's Inequality. For cases where $\Psi$ or $\Gamma$ is linear, our proof shows that the resulting bounds are essentially optimal, modulo the use of Hölder's inequality. As we shall see in Section 5, $\Psi$ and $\Gamma$ are linear when Massart's noise assumption holds.

Building upon these theorems, we proceed to derive finer $\mathcal{H}$-consistency bounds than existing ones.

## 4. Standard multi-class classification

We first apply our new tools to establish enhanced $\mathcal{H}$-consistency bounds in standard multi-class classification. We will consider the constrained losses (Lee et al., 2004), defined as

$$\Phi^{\text{cstnd}}(h, x, y) = \sum_{y' \neq y} \Phi\big(-h(x, y')\big) \text{ subject to } \sum_{y \in \mathcal{Y}} h(x, y) = 0, \tag{5}$$

where $\Phi$ is a non-increasing and non-negative function. We will specifically consider $\Phi(u) = e^{-u}$, $\Phi(u) = \max\{0, 1 - u\}$, and $\Phi(u) = (1 - u)^2 1_{u \leq 1}$, corresponding to the constrained exponential loss, constrained hinge loss, and constrained squared hinge loss, respectively. By applying Theorems 1 or 2, we obtain the following enhanced $\mathcal{H}$-consistency bounds. We say that a hypothesis set is symmetric if there exists a family $\mathcal{F}$ of functions $f$ mapping from $\mathcal{X}$ to $\mathbb{R}$ such that

$\{[h(x,1),\ldots,h(x,n+1)]\colon h \in \mathcal{H}\} = \{[f_1(x),\ldots,f_{n+1}(x)]\colon f_1,\ldots,f_{n+1} \in \mathcal{F}\}$, for any $x \in \mathcal{X}$. We say that a hypothesis set $\mathcal{H}$ is complete if for any $(x,y) \in \mathcal{X} \times \mathcal{Y}$, the set of scores generated by it spans across the real numbers: $\{h(x,y) \mid h \in \mathcal{H}\} = \mathbb{R}$.

**Theorem 5 (Enhanced $\mathcal{H}$-consistency bounds for constrained losses)** *Assume that $\mathcal{H}$ is symmetric and complete. Then, the following inequality holds for any hypothesis $h \in \mathcal{H}$:*

$$\mathcal{E}_{\ell_{0-1}}(h) - \mathcal{E}^*_{\ell_{0-1}}(\mathcal{H}) + \mathcal{M}_{\ell_{0-1}}(\mathcal{H}) \leq \Gamma\big(\mathcal{E}_{\Phi^{\mathrm{cstnd}}}(h) - \mathcal{E}^*_{\Phi^{\mathrm{cstnd}}}(\mathcal{H}) + \mathcal{M}_{\Phi^{\mathrm{cstnd}}}(\mathcal{H})\big),$$

*where* $\Gamma(x) = \dfrac{\sqrt{2}\,x^{\frac{1}{2}}}{\big(e^{\Lambda(h)}\big)^{\frac{1}{2}}}$ *for* $\Phi(u) = e^{-u}$, $\Gamma(x) = \dfrac{x}{1+\Lambda(h)}$ *for* $\Phi(u) = \max\{0, 1-u\}$, *and* $\Gamma(x) = \dfrac{x^{\frac{1}{2}}}{1+\Lambda(h)}$ *for* $\Phi(u) = (1-u)^2 1_{u\leq 1}$. *Additionally,* $\Lambda(h) = \inf_{x \in \mathcal{X}} \max_{y \in \mathcal{Y}} h(x,y)$.

The proof is included in Appendix C. These $\mathcal{H}$-consistency bounds are referred to as enhanced $\mathcal{H}$-consistency bounds because they incorporate a hypothesis-dependent quantity, $\Lambda(h)$, unlike the previous $\mathcal{H}$-consistency bounds derived for the constrained losses in (Awasthi et al., 2022b). Since $\sum_{y \in \mathcal{Y}} h(x,y) = 0$, there must be non-negative scores. Consequently, $\Lambda(h)$ must be greater than or equal to 0. Given that $\Gamma$ is non-decreasing, the $\mathcal{H}$-consistency bounds in Theorem 5 are finer than the previous ones, where $\Lambda(h)$ is replaced by zero.

## 5. Classification under low-noise conditions

The previous section demonstrated the usefulness of our new fundamental tools in deriving enhanced $\mathcal{H}$-consistency bounds within standard classification settings. In this section, we leverage them to establish novel $\mathcal{H}$-consistency bounds under low-noise conditions for both binary and multi-class classification problems.

### 5.1. Binary classification

Here, we first consider the binary classification setting under the Tsybakov noise condition (Mammen and Tsybakov, 1999), that is there exist $B > 0$ and $\alpha \in [0,1)$ such that

$$\forall t > 0, \quad \mathbb{P}[|\eta(x) - 1/2| \leq t] \leq B t^{\frac{\alpha}{1-\alpha}}.$$

Note that as $\alpha \to 1$, $t^{\frac{\alpha}{1-\alpha}} \to 0$, corresponding to Massart's noise condition. When $\alpha = 0$, the condition is void. This condition is equivalent to assuming the existence of a universal constant $c > 0$ and $\alpha \in [0,1)$ such that for all $h \in \mathcal{H}$, the following inequality holds (Bartlett et al., 2006):

$$\mathbb{E}\big[1_{\mathsf{h}(X)\neq\mathsf{h}^*(X)}\big] \leq c\Big[\mathcal{E}_{\ell^{\mathrm{bi}}_{0-1}}(h) - \mathcal{E}_{\ell^{\mathrm{bi}}_{0-1}}(h^*)\Big]^{\alpha}.$$

where $h^*$ is the Bayes-classifier. We also assume that there is no approximation error and that $\mathcal{M}_{\ell^{\mathrm{bi}}_{0-1}}(\mathcal{H}) = 0$. We refer to this as the *Tsybakov noise assumption* in binary classification.

**Theorem 6** *Consider a binary classification setting where the Tsybakov noise assumption holds. Assume that the following holds for all $h \in \mathcal{H}$ and $x \in \mathcal{X}$:* $\Delta\mathcal{C}_{\ell^{\mathrm{bi}}_{0-1},\mathcal{H}}(h,x) \leq \Gamma\big(\Delta\mathcal{C}_{\ell,\mathcal{H}}(h,x)\big)$, *with* $\Gamma(x) = x^{\frac{1}{s}}$, *for some $s \geq 1$. Then, for any $h \in \mathcal{H}$,*

$$\mathcal{E}_{\ell^{\mathrm{bi}}_{0-1}}(h) - \mathcal{E}^*_{\ell^{\mathrm{bi}}_{0-1}}(\mathcal{H}) \leq c^{\frac{s-1}{s-\alpha(s-1)}}\big[\mathcal{E}_\ell(h) - \mathcal{E}^*_\ell(\mathcal{H}) + \mathcal{M}_\ell(\mathcal{H})\big]^{\frac{1}{s-\alpha(s-1)}}.$$

| Loss functions | $\Phi$ | $\Gamma$ | $\mathcal{H}$-consistency bounds |
|---|---|---|---|
| Hinge | $\Phi_{\mathrm{hinge}}(u) = \max\{0, 1-u\}$ | $x^1$ | $\mathcal{E}_\ell(h) - \mathcal{E}_\ell^*(\mathcal{H}) + \mathcal{M}_\ell(\mathcal{H})$ |
| Logistic | $\Phi_{\mathrm{log}}(u) = \log(1+e^{-u})$ | $x^2$ | $c^{\frac{1}{2-\alpha}}\big[\mathcal{E}_\ell(h) - \mathcal{E}_\ell^*(\mathcal{H}) + \mathcal{M}_\ell(\mathcal{H})\big]^{\frac{1}{2-\alpha}}$ |
| Exponential | $\Phi_{\mathrm{exp}}(u) = e^{-u}$ | $x^2$ | $c^{\frac{1}{2-\alpha}}\big[\mathcal{E}_\ell(h) - \mathcal{E}_\ell^*(\mathcal{H}) + \mathcal{M}_\ell(\mathcal{H})\big]^{\frac{1}{2-\alpha}}$ |
| Squared-hinge | $\Phi_{\mathrm{sq-hinge}}(u) = (1-u)^2 1_{u\le 1}$ | $x^2$ | $c^{\frac{1}{2-\alpha}}\big[\mathcal{E}_\ell(h) - \mathcal{E}_\ell^*(\mathcal{H}) + \mathcal{M}_\ell(\mathcal{H})\big]^{\frac{1}{2-\alpha}}$ |
| Sigmoid | $\Phi_{\mathrm{sig}}(u) = 1 - \tanh(ku),\ k>0$ | $x^1$ | $\mathcal{E}_\ell(h) - \mathcal{E}_\ell^*(\mathcal{H}) + \mathcal{M}_\ell(\mathcal{H})$ |
| $\rho$-Margin | $\Phi_\rho(u) = \min\big\{1, \max\big\{0, 1-\frac{u}{\rho}\big\}\big\},\ \rho>0$ | $x^1$ | $\mathcal{E}_\ell(h) - \mathcal{E}_\ell^*(\mathcal{H}) + \mathcal{M}_\ell(\mathcal{H})$ |

Table 1: Examples of enhanced $\mathcal{H}$-consistency upper bounds under the Tsybakov noise assumption and with complete hypothesis sets, for margin-based loss functions $\ell(h,x,y) = \Phi(yh(x))$.

The theorem offers a substantially more favorable $\mathcal{H}$-consistency guarantee for binary classification. While standard $\mathcal{H}$-consistency bounds for smooth loss functions rely on a square-root dependency ($s = 2$), this work establishes a linear dependence when Massart's noise condition holds ($\alpha \to 1$), and an intermediate rate between linear and square-root for other values of $\alpha$ within the range $(0,1)$.

Our result is general and admits as special cases previous related bounds. In particular, setting $s = 2$ and $\alpha = 1$ recovers the $\mathcal{H}$-consistency bounds of (Awasthi et al., 2022a) under Massart's noise. Additionally, with $\mathcal{H} = \mathcal{H}_{\mathrm{all}}$, it recovers the excess bounds under the Tsybakov noise condition of (Bartlett et al., 2006), but with a more favorable factor of one instead of $2^{\frac{s}{s-\alpha(s-1)}}$, which is always greater than one. Table 1 illustrates several specific instances of our bounds for margin-based losses.

The proof is given in Appendix D.1. It consists of defining $\beta(h,x) = \frac{1_{h(x)\ne h^*(x)}+\epsilon}{\mathbb{E}_X\big[1_{h(x)\ne h^*(x)}+\epsilon\big]}$ for a fixed $\epsilon > 0$ and proving the inequality $\frac{\Delta\mathcal{C}_{\ell_{0-1}^{\mathrm{bi}},\mathcal{H}}(h,x)}{\beta(h,x)} \le \big(\alpha(h,x)\,\Delta\mathcal{C}_{\ell,\mathcal{H}}(h,x)\big)^{\frac{1}{s}}$, where $\alpha(h,x) = \mathbb{E}_X\big[1_{h(x)\ne h^*(x)} + \epsilon\big]^s$. The result then follows the application of our new tools Theorem 4. Note that our proof is novel and that previous general tools for deriving $\mathcal{H}$-consistency bounds in (Awasthi et al., 2022a,b; Zheng et al., 2023) cannot be applied here since $\alpha$ and $\beta$ are not constants.

## 5.2. Multi-class classification

The original definition of the Tsybakov noise (Mammen and Tsybakov, 1999) was given and analyzed in the binary classification setting. Here, we give a natural extension of this definition and analyze its properties in the general multi-class classification setting. We denote by $y_{\mathrm{max}} = \mathrm{argmax}_{y\in\mathcal{Y}} p(y \mid x)$. Define the minimal margin for a point $x \in \mathcal{X}$ as follows: $\gamma(x) = \mathbb{P}(y_{\mathrm{max}}|x) - \sup_{y\ne y_{\mathrm{max}}} \mathbb{P}(y|x)$. The Tsybakov noise model assumes that the probability of a small margin occurring is relatively low, that is there exist $B > 0$ and $\alpha \in [0,1)$ such that

$$\forall t > 0, \quad \mathbb{P}[\gamma(X) \le t] \le B t^{\frac{\alpha}{1-\alpha}}. \tag{6}$$

In the binary classification setting, where $\gamma(x) = 2\eta(x) - 1$, this recovers the condition described in Section 5.1. For $\alpha \to 1$, $t^{\frac{\alpha}{1-\alpha}} \to 0$, this corresponds to Massart's noise condition in multi-class classification. When $\alpha = 0$, the condition becomes void. Similar to the binary classification setting, we can establish an equivalence assumption for the Tsybakov noise model as follows. We denote the Bayes classifier by $h^*$.

**Lemma 7** *The Tsybakov noise assumption implies that there exists a constant $c$ such that the following inequalities hold for any $h \in \mathcal{H}$:*

$$\mathbb{E}[1_{\mathsf{h}(x) \neq \mathsf{h}^*(x)}] \leq c\,\mathbb{E}[\gamma(X)1_{\mathsf{h}(x) \neq \mathsf{h}^*(x)}]^\alpha \leq c[\mathcal{E}_{\ell_{0-1}}(h) - \mathcal{E}_{\ell_{0-1}}(h^*)]^\alpha.$$

**Lemma 8** *Assume that for any $h \in \mathcal{H}_{\text{all}}$, we have $\mathbb{P}[\mathsf{h}(X) \neq \mathsf{h}^*(X)] \leq c\,\mathbb{E}[\gamma(X)1_{\gamma(X) \leq t}]^\alpha$. Then, the Tsybakov noise condition holds, that is, there exists a constant $B > 0$, such that*

$$\forall t > 0, \quad \mathbb{P}[\gamma(X) \leq t] \leq Bt^{\frac{\alpha}{1-\alpha}}.$$

The proofs of Lemma 7 and Lemma 8 are included in Appendix D.2. To the best of our knowledge, there are no previous results that formally analyze these properties of the Tsybakov noise in the general multi-class classification setting, although the similar result in the binary setting is well-known. Next, we assume that there exists a universal constant $c > 0$ and $\alpha \in [0, 1)$ such that for all $h \in \mathcal{H}$, the following Tsybakov noise inequality holds:

$$\mathbb{E}[1_{\mathsf{h}(x) \neq \mathsf{h}^*(x)}] \leq c[\mathcal{E}_{\ell_{0-1}}(h) - \mathcal{E}_{\ell_{0-1}}(h^*)]^\alpha. \tag{7}$$

where $h^*$ is the Bayes-classifier. We also assume that there is no approximation error and that $\mathcal{M}_{\ell_{0-1}}(\mathcal{H}) = 0$. We refer to this as the *Tsybakov noise assumption* in multi-class classification.

**Theorem 9** *Consider a multi-class classification setting where the Tsybakov noise assumption holds. Assume that the following holds for all $h \in \mathcal{H}$ and $x \in \mathcal{X}$: $\Delta\mathcal{C}_{\ell_{0-1},\mathcal{H}}(h, x) \leq \Gamma\big(\Delta\mathcal{C}_{\ell,\mathcal{H}}(h, x)\big)$, with $\Gamma(x) = x^{\frac{1}{s}}$, for some $s \geq 1$. Then, for any $h \in \mathcal{H}$,*

$$\mathcal{E}_{\ell_{0-1}}(h) - \mathcal{E}_{\ell_{0-1}}^*(\mathcal{H}) \leq c^{\frac{s-1}{s-\alpha(s-1)}}\big[\mathcal{E}_\ell(h) - \mathcal{E}_\ell^*(\mathcal{H}) + \mathcal{M}_\ell(\mathcal{H})\big]^{\frac{1}{s-\alpha(s-1)}}.$$

To our knowledge, these are the first multi-class classification $\mathcal{H}$-consistency bounds, and even excess error bounds (a special case where $\mathcal{H} = \mathcal{H}_{\text{all}}$) established under the Tsybakov noise assumption. Here too, this theorem offers a significantly improved $\mathcal{H}$-consistency guarantee for multi-class classification. For smooth loss functions, standard $\mathcal{H}$-consistency bounds rely on a square-root dependence ($s = 2$). This dependence is improved to a linear rate when the Massart noise condition holds ($\alpha \to 1$), or to an intermediate rate between linear and square-root for other values of $\alpha$ within the range $(0, 1)$. The proof is given in Appendix D.3. Illustrative examples of these bounds for constrained losses and comp-sum losses are presented in Tables 2 and 3.

## 6. Bipartite ranking

In preceding sections, we demonstrated how our new tools enable the derivation of enhanced $\mathcal{H}$-consistency bounds in various classification scenarios: standard multi-class classification and low-noise regimes of both binary and multi-class classification. Here, we extend the applicability of our refined tools to the bipartite ranking setting. We illustrate how they facilitate the establishment of more favorable $\mathcal{H}$-consistency bounds for classification surrogate losses $\ell_\Phi$ with respect to the bipartite ranking surrogate losses $\mathsf{L}_\Phi$. The loss functions $\ell_\Phi$ and $\mathsf{L}_\Phi$ are defined as follows:

$$\ell_\Phi(h, x, y) = \Phi(yh(x)), \quad \mathsf{L}_\Phi(h, x, x', y, y') = \Phi\Big(\frac{(y-y')(h(x)-h(x'))}{2}\Big)1_{y \neq y'},$$

| Loss functions | $\ell$ | $\Gamma$ | $\mathcal{H}$-consistency bounds |
|---|---|---|---|
| Constrained hinge | $\sum_{y'\neq y}\Phi_{\mathrm{hinge}}(-h(x,y'))$ | $x^1$ | $\mathcal{E}_\ell(h)-\mathcal{E}_\ell^*(\mathcal{H})+\mathcal{M}_\ell(\mathcal{H})$ |
| Constrained exponential | $\sum_{y'\neq y}\Phi_{\exp}(-h(x,y'))$ | $x^2$ | $c^{\frac{1}{2-\alpha}}\big[\mathcal{E}_\ell(h)-\mathcal{E}_\ell^*(\mathcal{H})+\mathcal{M}_\ell(\mathcal{H})\big]^{\frac{1}{2-\alpha}}$ |
| Constrained squared-hinge | $\sum_{y'\neq y}\Phi_{\mathrm{sq-hinge}}(-h(x,y'))$ | $x^2$ | $c^{\frac{1}{2-\alpha}}\big[\mathcal{E}_\ell(h)-\mathcal{E}_\ell^*(\mathcal{H})+\mathcal{M}_\ell(\mathcal{H})\big]^{\frac{1}{2-\alpha}}$ |
| Constrained $\rho$-margin | $\sum_{y'\neq y}\Phi_\rho(-h(x,y'))$ | $x^1$ | $\mathcal{E}_\ell(h)-\mathcal{E}_\ell^*(\mathcal{H})+\mathcal{M}_\ell(\mathcal{H})$ |

Table 2: Examples of enhanced $\mathcal{H}$-consistency bounds under the Tsybakov noise assumption and with symmetric and complete hypothesis sets, as provided by Theorem 9, for constrained losses $\ell(h,x,y)=\Phi^{\mathrm{cstnd}}(h,x,y)=\sum_{y'\neq y}\Phi(-h(x,y'))$ subject to $\sum_{y\in\mathcal{Y}}h(x,y)=0$ (with only the surrogate portion displayed).

| Loss functions | $\ell$ | $\Gamma$ | $\mathcal{H}$-consistency bounds |
|---|---|---|---|
| Sum exponential | $\sum_{y'\neq y}e^{h(x,y')-h(x,y)}$ | $x^1$ | $\mathcal{E}_\ell(h)-\mathcal{E}_\ell^*(\mathcal{H})+\mathcal{M}_\ell(\mathcal{H})$ |
| Multinomial logistic | $-\log\Big(\frac{e^{h(x,y)}}{\sum_{y'\in\mathcal{Y}}e^{h(x,y')}}\Big)$ | $x^2$ | $c^{\frac{1}{2-\alpha}}\big[\mathcal{E}_\ell(h)-\mathcal{E}_\ell^*(\mathcal{H})+\mathcal{M}_\ell(\mathcal{H})\big]^{\frac{1}{2-\alpha}}$ |
| Generalized cross-entropy | $\frac{1}{\alpha}\Big[1-\Big[\frac{e^{h(x,y)}}{\sum_{y'\in\mathcal{Y}}e^{h(x,y')}}\Big]^\alpha\Big]$ | $x^2$ | $c^{\frac{1}{2-\alpha}}\big[\mathcal{E}_\ell(h)-\mathcal{E}_\ell^*(\mathcal{H})+\mathcal{M}_\ell(\mathcal{H})\big]^{\frac{1}{2-\alpha}}$ |
| Mean absolute error | $1-\frac{e^{h(x,y)}}{\sum_{y'\in\mathcal{Y}}e^{h(x,y')}}$ | $x^1$ | $\mathcal{E}_\ell(h)-\mathcal{E}_\ell^*(\mathcal{H})+\mathcal{M}_\ell(\mathcal{H})$ |

Table 3: Examples of enhanced $\mathcal{H}$-consistency bounds under the Tsybakov noise assumption and with symmetric and complete hypothesis sets, as provided by Theorem 9, for comp-sum losses (with only the surrogate portion displayed).

where $\Phi$ is a non-negative and non-increasing function. We will say that $\ell_\Phi$ admits an $\mathcal{H}$-consistency bound with respect to $\mathsf{L}_\Phi$, if there exists a concave function $\Gamma\colon\mathbb{R}_+\to\mathbb{R}_+$ with $\Gamma(0)=0$ such that the following inequality holds:

$$\mathcal{E}_{\mathsf{L}_\Phi}(h)-\mathcal{E}_{\mathsf{L}_\Phi}^*(\mathcal{H})+\mathcal{M}_{\mathsf{L}_\Phi}(\mathcal{H})\le\Gamma\big(\mathcal{E}_{\ell_\Phi}(h)-\mathcal{E}_{\ell_\Phi}^*(\mathcal{H})+\mathcal{M}_{\ell_\Phi}(\mathcal{H})\big),$$

where $\mathcal{M}_{\ell_\Phi}(\mathcal{H})=\mathcal{E}_{\ell_\Phi}^*(\mathcal{H})-\mathbb{E}_X\big[\mathcal{C}_{\ell_\Phi}^*(\mathcal{H},x)\big]$ and $\mathcal{M}_{\mathsf{L}_\Phi}(\mathcal{H})=\mathcal{E}_{\mathsf{L}_\Phi}^*(\mathcal{H})-\mathbb{E}_{(x,x')}\big[\overline{\mathcal{C}}_{\mathsf{L}_\Phi}^*(\mathcal{H},x,x')\big]$ represent the minimizability gaps for $\ell_\Phi$ and $\mathsf{L}_\Phi$, respectively.

## 6.1. Fundamental tools for bipartite ranking

We first extend our new fundamental tools to the bipartite ranking setting.

**Theorem 10** *Assume that there exist two concave functions $\Gamma_1\colon\mathbb{R}_+\to\mathbb{R}$ and $\Gamma_2\colon\mathbb{R}_+\to\mathbb{R}$, and two positive functions $\alpha_1\colon\mathcal{H}\times\mathcal{X}\to\mathbb{R}_+^*$ and $\alpha_2\colon\mathcal{H}\times\mathcal{X}\to\mathbb{R}_+^*$ with $\mathbb{E}_{x\in\mathcal{X}}[\alpha_1(h,x)]<+\infty$ and $\mathbb{E}_{x\in\mathcal{X}}[\alpha_2(h,x)]<+\infty$ for all $h\in\mathcal{H}$ such that the following holds for all $h\in\mathcal{H}$ and $(x,x')\in\mathcal{X}\times\mathcal{X}$: $\Delta\overline{\mathcal{C}}_{\mathsf{L},\mathcal{H}}(h,x,x')\le\Gamma_1\big(\alpha_1(h,x')\,\Delta\mathcal{C}_{\ell,\mathcal{H}}(h,x)\big)+\Gamma_2\big(\alpha_2(h,x)\,\Delta\mathcal{C}_{\ell,\mathcal{H}}(h,x')\big)$. Then, the following inequality holds for any hypothesis $h\in\mathcal{H}$:*

$$\mathcal{E}_{\mathsf{L}}(h)-\mathcal{E}_{\mathsf{L}}^*(\mathcal{H})+\mathcal{M}_{\mathsf{L}}(\mathcal{H})\le\Gamma_1(\gamma_1(h)\mathsf{D}_\ell(h))+\Gamma_2(\gamma_2(h)\mathsf{D}_\ell(h)).$$

*with $\gamma_1(h)=\mathbb{E}_{x\in\mathcal{X}}[\alpha_1(h,x)]$, $\gamma_2(h)=\mathbb{E}_{x\in\mathcal{X}}[\alpha_2(h,x)]$, and $\mathsf{D}_\ell(h)=\mathcal{E}_\ell(h)-\mathcal{E}_\ell^*(\mathcal{H})+\mathcal{M}_\ell(\mathcal{H})$.*

The proof, detailed in Appendix E.1, leverages the fact that in the bipartite ranking setting, two pairs $(x, y)$ and $(x', y')$ are drawn i.i.d. according to the distribution $\mathcal{D}$. As in the classification setting, Theorem 10 is a fundamental tool for establishing enhanced $\mathcal{H}$-consistency bounds. This is achieved incorporating the additional terms $\alpha_1$ and $\alpha_2$, which can depend on both the hypothesis $h$ and the instances $x$ or $x'$, thereby offering greater flexibility.

Note that such enhanced $\mathcal{H}$-consistency bounds are meaningful only when $\Gamma_1(0) + \Gamma_2(0) = 0$. This ensures that when the minimizability gaps vanish (e.g., in the case where $\mathcal{H} = \mathcal{H}_{\mathrm{all}}$ or in more generally realizable cases), the estimation error of classification losses $\mathcal{E}_\ell(h) - \mathcal{E}_\ell^*(\mathcal{H})$ is zero implies that the estimation error of bipartite ranking losses $\mathcal{E}_{\mathsf{L}}(h) - \mathcal{E}_{\mathsf{L}}^*(\mathcal{H})$ is also zero. This requires that there exists $\Gamma_1$ and $\Gamma_2$ such that $\Gamma_1(0) + \Gamma_2(0) = 0$ and $\Delta\overline{\mathcal{C}}_{\mathsf{L},\mathcal{H}}(h, x, x') \leq \Gamma_1\big(\alpha_1(h, x')\,\Delta\mathcal{C}_{\ell,\mathcal{H}}(h, x)\big) + \Gamma_2\big(\alpha_2(h, x)\,\Delta\mathcal{C}_{\ell,\mathcal{H}}(h, x')\big)$, for all $h \in \mathcal{H}$ and $(x, x') \in \mathcal{X} \times \mathcal{X}$. Note that a necessary condition for this requirement is *calibration*: we say that a classification loss $\ell$ is *calibrated* with respect to a bipartite ranking loss $\mathsf{L}$, if for all $h \in \mathcal{H}_{\mathrm{all}}$ and $(x, x') \in \mathcal{X} \times \mathcal{X}$:

$$\Delta\mathcal{C}_{\ell,\mathcal{H}_{\mathrm{all}}}(h, x) = 0 \text{ and } \Delta\mathcal{C}_{\ell,\mathcal{H}_{\mathrm{all}}}(h, x') = 0 \implies \Delta\overline{\mathcal{C}}_{\mathsf{L},\mathcal{H}_{\mathrm{all}}}(h, x, x') = 0.$$

We now introduce a family of auxiliary functions that are differentiable and that admit a property facilitating the calibration between $\ell_\Phi$ and $\mathsf{L}_\Phi$.

**Theorem 11** *Assume that $\Phi$ is convex and differentiable, and satisfies $\Phi'(t) < 0$ for all $t \in \mathbb{R}$, and $\frac{\Phi'(t)}{\Phi'(-t)} = e^{-\nu t}$ for some $\nu > 0$. Then, $\ell_\Phi$ is calibrated with respect to $\mathsf{L}_\Phi$.*

The proof can be found in Appendix E.3. Theorem 11 identifies a family of functions $\Phi$ for which $\ell_\Phi$ is calibrated with respect to $\mathsf{L}_\Phi$. This inclues the exponential loss and the logistic loss, which fulfill the properties outlined in Theorem 11. For the exponential loss, $\Phi(u) = \Phi_{\exp}(u) = e^{-u}$, we have $\frac{\Phi'_{\exp}(t)}{\Phi'_{\exp}(-t)} = \frac{-e^{-t}}{-e^t} = e^{-2t}$. Similarly, for the logistic loss, $\Phi(u) = \Phi_{\log}(u) = \log(1 + e^{-u})$, we have $\frac{\Phi'_{\log}(t)}{\Phi'_{\log}(-t)} = \frac{-\frac{1}{e^t+1}}{-\frac{1}{e^{-t}+1}} = e^{-t}$. In the next sections, we will prove $\mathcal{H}$-consistency bounds in these two cases.

## 6.2. Exponential loss

We first consider the exponential loss, where $\Phi(u) = \Phi_{\exp}(u) = e^{-u}$. In the bipartite ranking setting, a hypothesis set $\mathcal{H}$ is said to be *complete* if for any $x \in \mathcal{X}$, $\{h(x) : h \in \mathcal{H}\}$ spans $\mathbb{R}$.

**Theorem 12** *Assume that $\mathcal{H}$ is complete. Then, the following inequality holds for the exponential loss $\Phi_{\exp}$:*

$$\Delta\overline{\mathcal{C}}_{\mathsf{L}_{\Phi_{\exp}},\mathcal{H}}(h, x, x') \leq \mathcal{C}_{\ell_{\Phi_{\exp}}}(h, x')\,\Delta\mathcal{C}_{\ell_{\Phi_{\exp}},\mathcal{H}}(h, x) + \mathcal{C}_{\ell_{\Phi_{\exp}}}(h, x)\,\Delta\mathcal{C}_{\ell_{\Phi_{\exp}},\mathcal{H}}(h, x').$$

*Additionally, for any hypothesis $h \in \mathcal{H}$, we have*

$$\mathcal{E}_{\mathsf{L}_{\Phi_{\exp}}}(h) - \mathcal{E}_{\mathsf{L}_{\Phi_{\exp}}}^*(\mathcal{H}) + \mathcal{M}_{\mathsf{L}_{\Phi_{\exp}}}(\mathcal{H}) \leq 2\mathcal{E}_{\ell_{\Phi_{\exp}}}(h)\Big(\mathcal{E}_{\ell_{\Phi_{\exp}}}(h) - \mathcal{E}_{\ell_{\Phi_{\exp}}}^*(\mathcal{H}) + \mathcal{M}_{\ell_{\Phi_{\exp}}}(\mathcal{H})\Big).$$

See Appendix E.2 for the proof. The proof leverages our new tool, Theorem 10, in conjunction with the specific form of the conditional regrets for the exponential function and the convexity of squared function. This result is remarkable since it directly bounds the estimation error of the

RankBoost loss function by that of AdaBoost. The observation that AdaBoost often exhibits favorable ranking accuracy, often approaching that of RankBoost, was first highlighted by Cortes and Mohri (2003). Later, Rudin et al. (2005) introduced a coordinate descent version of RankBoost and demonstrated that, when incorporating the constant weak classifier, AdaBoost asymptotically achieves the same ranking accuracy as coordinate descent RankBoost.

Here, we present a stronger non-asymptotic result for the estimation losses of these algorithms. We show that when the estimation error of the AdaBoost predictor $h$ is reduced to $\epsilon$, the corresponding RankBoost loss is bounded by $2\mathcal{E}_{\ell_{\Phi_{\exp}}}(h)\left(\epsilon + \mathcal{M}_{\ell_{\Phi_{\exp}}}(\mathcal{H})\right) - \mathcal{M}_{\mathsf{L}_{\Phi_{\exp}}}(\mathcal{H})$. This provides a stronger guarantee for the ranking quality of AdaBoost. In the nearly realizable case, where minimizability gaps are negligible, this upper bound approximates to $2\mathcal{E}_{\ell_{\Phi_{\exp}}}(h)\epsilon$, aligning with the results of Gao and Zhou (2015) for excess errors, where $\mathcal{H}$ is assumed to be the family of all measurable functions.

### 6.3. Logistic loss

Here, we consider the logistic loss, where $\Phi(u) = \Phi_{\log}(u) = \log(1 + e^{-u})$.

**Theorem 13** *Assume that $\mathcal{H}$ is complete. For any $x$, define $u(x) = \max\{\eta(x), 1 - \eta(x)\}$. Then, the following inequality holds for the logistic loss $\Phi_{\log}$:*

$$\Delta\overline{\mathcal{C}}_{\mathsf{L}_{\Phi_{\log}},\mathcal{H}}(h, x, x') \le u(x')\Delta\mathcal{C}_{\ell_{\Phi_{\log}},\mathcal{H}}(h, x) + u(x)\Delta\mathcal{C}_{\ell_{\Phi_{\log}},\mathcal{H}}(h, x').$$

*Furthermore, for any hypothesis $h \in \mathcal{H}$, we have*

$$\mathcal{E}_{\mathsf{L}_{\Phi_{\log}}}(h) - \mathcal{E}^*_{\mathsf{L}_{\Phi_{\log}}}(\mathcal{H}) + \mathcal{M}_{\mathsf{L}_{\Phi_{\log}}}(\mathcal{H}) \le 2\,\mathbb{E}[u(X)]\left(\mathcal{E}_{\ell_{\Phi_{\log}}}(h) - \mathcal{E}^*_{\ell_{\Phi_{\log}}}(\mathcal{H}) + \mathcal{M}_{\ell_{\Phi_{\log}}}(\mathcal{H})\right).$$

Note that the term $\mathbb{E}[u(X)]$ can be expressed as $1 - \mathbb{E}[\min\{\eta(X), (1 - \eta(X))\}]$, and coincides with the accuracy of the Bayes classifier. In particular, in the deterministic case, we have $\mathbb{E}[u(X)] = 1$. The proof is given in Appendix E.4. In the first part of the proof, we establish and leverage the sub-additivity of $\Phi_{\log}$: $\Phi_{\log}(h - h') \le \Phi_{\log}(h) + \Phi_{\log}(-h')$, to derive an upper bound for $\Delta\overline{\mathcal{C}}_{\mathsf{L}_{\Phi_{\log}},\mathcal{H}}(h, x, x')$ in terms of $\Delta\mathcal{C}_{\ell_{\Phi_{\log}},\mathcal{H}}(h, x)$ and $\Delta\mathcal{C}_{\ell_{\Phi_{\log}},\mathcal{H}}(h, x')$. Next, we apply our new tool, Theorem 10, with $\alpha_1(h, x') = \max\{\eta(x'), 1 - \eta(x')\}$ and $\alpha_2(h, x) = \max\{\eta(x), 1 - \eta(x)\}$.

Both our result and its proof are entirely novel. Significantly, this result implies a parallel finding for logistic regression analogous to that of AdaBoost: If $h$ is the predictor obtained by minimizing the logistic loss estimation error to $\epsilon$, then the $\mathsf{L}_{\Phi_{\log}}$-estimation loss of $h$ for ranking is bounded above by $2\,\mathbb{E}[u(X)](\epsilon + \mathcal{M}_{\ell_{\Phi_{\log}}}(\mathcal{H})) - \mathcal{M}_{\mathsf{L}_{\Phi_{\log}}}(\mathcal{H})$. When minimizability gaps are small, such as in realizable cases, this bound further simplifies to $2\,\mathbb{E}[u(X)]\epsilon$, suggesting a favorable ranking property for logistic regression.

This result is surprising, as the favorable ranking property of AdaBoost and its connection to RankBoost were thought to stem from the specific properties of the exponential loss, particularly its morphism property, which directly links the loss functions of AdaBoost and RankBoost. This direct connection does not exist for the logistic loss, making our proof and result particularly remarkable. In both cases, our new tools facilitated the derivation of non-trivial inequalities where the factor plays a crucial role. The exploration of enhanced $\mathcal{H}$-consistency bounds for other functions $\Phi$ is an interesting question for future research that we have initiated. In the next section, we prove negative results for the hinge loss.

### 6.4. Hinge loss

The hinge loss $\ell_{\Phi_{\mathrm{hinge}}}$ is the loss function minimized by the support vector machines (SVM) (Cortes and Vapnik, 1995) and $\mathsf{L}_{\Phi_{\mathrm{hinge}}}$ is the loss function optimized by the RankSVM algorithm (Joachims, 2002) . However, the relationships observed for AdaBoost and RankBoost, or Logistic Regression and its ranking counterpart, do not hold here. Instead, we present the following two negative results.

**Theorem 14** *For the hinge loss, $\ell_{\Phi_{\mathrm{hinge}}}$ is not calibrated with respect to $\mathsf{L}_{\Phi_{\mathrm{hinge}}}$.*

**Theorem 15 (Negative result for hinge losses)** *Assume that $\mathcal{H}$ contains the constant function $1$. For the hinge loss, if there exists a function pair $(\Gamma_1, \Gamma_2)$ such that the following holds for all $h \in \mathcal{H}$ and $(x, x') \in \mathcal{X} \times \mathcal{X}$, with some positive functions $\alpha_1 \colon \mathcal{H} \times \mathcal{X} \to \mathbb{R}_+^*$ and $\alpha_2 \colon \mathcal{H} \times \mathcal{X} \to \mathbb{R}_+^*$:*

$$\Delta\overline{\mathcal{C}}_{\mathsf{L}_{\Phi_{\mathrm{hinge}}}, \mathcal{H}}(h, x, x') \le \Gamma_1\Big(\alpha_1(h, x')\,\Delta\mathcal{C}_{\ell_{\Phi_{\mathrm{hinge}}}, \mathcal{H}}(h, x)\Big) + \Gamma_2\Big(\alpha_2(h, x)\,\Delta\mathcal{C}_{\ell_{\Phi_{\mathrm{hinge}}}, \mathcal{H}}(h, x')\Big),$$

*then, we have $\Gamma_1(0) + \Gamma_2(0) \ge \frac{1}{2}$.*

See Appendix E.5 for the proof. Theorem 15 implies that there are no meaningful $\mathcal{H}$-consistency bounds for $\ell_{\Phi_{\mathrm{hinge}}}$ with respect to $\mathsf{L}_{\Phi_{\mathrm{hinge}}}$ with common hypothesis sets. In Appendix F, we show that all our derived enhanced $\mathcal{H}$-consistency bounds can be used to provide novel enhanced generalization bounds in their respective settings.

## 7. Discussion

**Role of non-constant factors**. One advantage of our enhanced $\mathcal{H}$-consistency bounds is their ability to incorporate non-constant factors that reflect both the data distribution and the predictor. For example, in the bounds with respect to the exponential loss in bipartite ranking, the non-constant factor is the generalization error of the AdaBoost-loss predictor. This means that the rate of the bound becomes more favorable as the predictor's performance approaches that of the best-in-class predictor. The best rate depends on the data distribution, as does the best-in-class generalization error. Similarly, in the enhanced $\mathcal{H}$-consistency bounds with respect to the logistic loss in bipartite ranking, the non-constant factor is the accuracy of the Bayes classifier. This means that the rate of the bound depends on the data distribution. In particular, in the deterministic case, the accuracy of the Bayes classifier is one.

**Applicability and significance of our new tools**. Our new fundamental tools enable the derivation of more favorable guarantees in various scenarios that (i) better leverage key distributional properties; (ii) establish connections between existing algorithms; and (iii) can lead to more favorable algorithms in other scenarios. For (i), an example is our more favorable $\mathcal{H}$-consistency bounds under low-noise conditions for both binary and multi-class classification problems, with a linear rate when the Massart noise condition holds and an intermediate rate between linear and square-root for other values of $\alpha$ under the Tsybakov noise assumption. For (ii), an example of this is our enhanced $\mathcal{H}$-consistency bounds in bipartite ranking, which provide a theoretical explanation for the empirical observation that AdaBoost tends to perform well in ranking tasks. A similar insight holds for the logistic loss. For (iii), our new tools can also be applied to comp-sum losses, and the enhanced bounds may contribute to the development of more robust adversarial algorithms. This is similar

to the improvements in adversarial robustness achieved in (Mao et al., 2023f), which presents an interesting direction for future research.

**Connection to existing bipartite ranking results**. Prior work, such as (Kotlowski et al., 2011) and (Agarwal, 2014), has established the Bayes-consistency of several classification surrogate losses with respect to bipartite misranking loss. Specifically, Kotlowski et al. (2011) examined the exponential and logistic surrogate losses, while Agarwal (2014) explored a broader class of proper (composite) classification surrogate losses.

In contrast, our work focuses on establishing enhanced $\mathcal{H}$-consistency bounds for classification surrogate losses with respect to surrogate bipartite misranking losses. For instance, we prove $\mathcal{H}$-consistency bounds for the classification exponential loss (used in AdaBoost) with respect to the bipartite misranking exponential loss (used in RankBoost). These are in some sense stronger results since, combined with the standard consistency of bipartite misranking surrogate losses with respect to the bipartite misranking loss, our results imply the consistency of classification surrogate losses with respect to the bipartite misranking loss. Moreover, our contributions go beyond Bayes-consistency by providing more informative $\mathcal{H}$-consistency bounds. An interesting future direction is to extend our enhanced $\mathcal{H}$-consistency bounds to more general strongly proper losses considered in (Agarwal, 2014) with respect to their corresponding bipartite misranking counterparts.

**Faster rates compared to previous work**. Recent work by Mao et al. (2024a) shows that for any smooth surrogate loss in binary and multi-class classification, $\Gamma(\epsilon)$ behaves as $\sqrt{\epsilon}$ near zero. However, our analysis demonstrates that under specific noise conditions (distributional assumptions), $\mathcal{H}$-consistency bounds can achieve significantly improved rates, even approaching near-linear behavior in limiting cases. It is important to emphasize that the bounds derived in (Mao et al., 2024a) are not incorrect and do not contradict our results. They are worst-case results that hold universally for *any* distribution. Specifically, their bounds rely on a fixed convex function $\Psi$ or concave function $\Gamma$, which is independent of both the distribution and the hypothesis. In contrast, our framework introduces the auxiliary functions $\alpha$ and $\beta$, which enable the derivation of refined bounds incorporating a non-constant factor $\gamma$ that adapts to both the data distribution and the predictor $h$.

Remarkably, under Tsybakov noise conditions, we can derive more favorable $\mathcal{H}$-consistency bounds (See Theorems 6 and 9, and the subsequent discussion) with better exponents. In the proof, we choose functions $\alpha$ and $\beta$ that depend on the input, the predictor $h$, and the best predictor $h^*$. For example, $\beta(h, x)$ measures the disagreement of $h$ and $h^*$ on $x$, modulo a small constant $\epsilon$.

## 8. Conclusion

We introduced novel tools for deriving enhanced $\mathcal{H}$-consistency bounds in various learning settings, including multi-class classification, low-noise regimes, and bipartite ranking. Remarkably, we established substantially more favorable guarantees for several settings and demonstrated unexpected connections between classification and bipartite ranking performances for the exponential and logistic losses. Our tools are likely to be useful in the analysis of $\mathcal{H}$-consistency bounds for a wide range of other scenarios.

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

# Contents of Appendix

## Appendix A.  Related work

Bayes-consistency has been well studied in a wide range of learning scenarios, including binary classification (Zhang, 2004a; Bartlett et al., 2006; Steinwart, 2007; Mohri et al., 2018), multi-class classification (Zhang, 2004b; Tewari and Bartlett, 2007; Ramaswamy and Agarwal, 2012; Ramaswamy et al., 2014; Narasimhan et al., 2015; Agarwal and Agarwal, 2015; Williamson et al., 2016; Ramaswamy and Agarwal, 2016; Chen and Sun, 2006; Chen and Xiang, 2006; Liu, 2007; Dogan et al., 2016; Wang and Scott, 2020, 2023, 2024), multi-label learning (Gao and Zhou, 2011; Dembczynski et al., 2012; Koyejo et al., 2015; Zhang et al., 2020), learning with rejection (Ramaswamy et al., 2015b; Cortes et al., 2016a,b, 2023; Ni et al., 2019; Cao et al., 2022), learning to defer (Mozannar and Sontag, 2020; Verma and Nalisnick, 2022; Cao et al., 2023), ranking (Ravikumar et al., 2011; Ramaswamy et al., 2013; Agarwal, 2014; Gao and Zhou, 2015; Uematsu and Lee, 2017), cost sensitive learning (Pires et al., 2013; Pires and Szepesvári, 2016), structured prediction (Ciliberto et al., 2016; Osokin et al., 2017; Blondel, 2019), general embedding framework (Finocchiaro et al., 2020; Frongillo and Waggoner, 2021; Finocchiaro et al., 2021, 2022; Nueve et al., 2024), Top-$k$ classification (Lapin et al., 2015; Yang and Koyejo, 2020; Thilagar et al., 2022), hierarchical classification (Ramaswamy et al., 2015a; Cao et al., 2024), ordinal regression (Pedregosa et al., 2017), and learning from noisy labels (Natarajan et al., 2013; Scott et al., 2013; Menon et al., 2015; Liu and Tao, 2015; Patrini et al., 2016; Liu and Guo, 2020; Zhang et al., 2021, 2022; Zhang and Agarwal, 2024). However, this classical notion has significant limitations since it only holds asymptotically and for the impractical set of all measurable functions. Thus, it fails to provide guarantees for real-world scenarios where learning is restricted to specific hypothesis sets, such as linear models or neural networks. In fact, Bayes-consistency does not always translate into superior performance, as highlighted by Long and Servedio (2013) (see also (Zhang and Agarwal, 2020)).

Awasthi et al. (2022a) proposed the key notion of $\mathcal{H}$-consistency bounds for binary classification. These novel non-asymptotic learning guarantees for binary classification account for the hypothesis set $\mathcal{H}$ adopted and are more significant and informative than existing Bayes-consistency guarantees. They provided general tools for deriving such bounds and used them to establish a series of $\mathcal{H}$-consistency bounds in both standard binary classification and binary classification under Massart's noise condition. Awasthi et al. (2022b) and Zheng et al. (2023) further generalized those general tools to standard multi-class classification and used them to establish multi-class $\mathcal{H}$-consistency bounds. Specifically, Awasthi et al. (2022b) presented a comprehensive analysis of $\mathcal{H}$-consistency bounds for the three most commonly used families of multi-class surrogate losses: *max losses* (Crammer and Singer, 2001), *sum losses* (Weston and Watkins, 1998), and *constrained losses* (Lee et al., 2004). They showed negative results for max losses, while providing positive results for sum losses and constrained losses. Additionally, Zheng et al. (2023) used these general tools in multi-class classification to derive $\mathcal{H}$-consistency bounds for the (multinomial) logistic loss (Verhulst, 1838, 1845; Berkson, 1944, 1951). Meanwhile, Mao et al. (2023f) presented a theoretical analysis of $\mathcal{H}$-consistency bounds for a broader family of loss functions, termed *comp-sum losses*, which includes sum losses and cross-entropy (or logistic loss) as special cases, and also includes generalized cross-entropy (Zhang and Sabuncu, 2018), mean absolute error (Ghosh et al., 2017), and other cross-entropy-like loss functions. In all these works, determining whether $\mathcal{H}$-consistency bounds hold and deriving these bounds have required specific proofs and analyses for each surrogate loss. Mao et al. (2023b) complemented these efforts by providing both a general characterization and an extension of $\mathcal{H}$-consistency bounds for multi-class classification, based on the error trans-

formation functions they defined for comp-sum losses and constrained losses. Recently, Mao et al. (2024a) further applied these error transformations to characterize the general behavior of these bounds, showing that the universal growth rate of $\mathcal{H}$-consistency bounds for smooth surrogate losses in both binary and multi-class classification is square-root. $\mathcal{H}$-consistency bounds have also been studied in other learning scenarios including pairwise ranking (Mao et al., 2023c,d), learning with rejection (Mao et al., 2024d,c; Mohri et al., 2024), learning to defer (Mao et al., 2023a, 2024b,h,g), top-$k$ classification (Cortes et al., 2024), adversarial robustness (Awasthi et al., 2021a,b, 2023a,b; Mao et al., 2023f), multi-label learning (Mao et al., 2024f), bounded regression (Mao et al., 2024e), and structured prediction (Mao et al., 2023e).

All previous bounds in the aforementioned work were derived under the condition that a lower bound of the surrogate loss conditional regret is given as a convex function of the target conditional regret, without non-constant factors depending on the predictor or input instance. In this work, we relax this condition and present a general framework for establishing enhanced $\mathcal{H}$-consistency bounds based on more general inequalities relating conditional regrets, leading to finer and more favorable $\mathcal{H}$-consistency bounds.

## Appendix B. Proof of new fundamental tools (Theorem 1, Theorem 2, Theorem 3 and Theorem 4)

**Theorem 1** *Assume that there exist a convex function* $\Psi\colon\mathbb{R}_+ \to \mathbb{R}$ *and two positive functions* $\alpha\colon\mathcal{H}\times\mathcal{X}\to\mathbb{R}_+^*$ *and* $\beta\colon\mathcal{H}\times\mathcal{X}\to\mathbb{R}_+^*$ *with* $\sup_{x\in\mathcal{X}}\alpha(h,x)<+\infty$ *and* $\mathbb{E}_{x\in\mathcal{X}}[\beta(h,x)]=1$ *for all* $h\in\mathcal{H}$ *such that the following holds for all* $h\in\mathcal{H}$ *and* $x\in\mathcal{X}$: $\Psi\left(\frac{\Delta\mathcal{C}_{\ell_2,\mathcal{H}}(h,x)}{\beta(h,x)}\right) \le \alpha(h,x)\,\Delta\mathcal{C}_{\ell_1,\mathcal{H}}(h,x)$. *Then, the following inequality holds for any hypothesis* $h\in\mathcal{H}$:

$$\Psi\big(\mathcal{E}_{\ell_2}(h) - \mathcal{E}_{\ell_2}^*(\mathcal{H}) + \mathcal{M}_{\ell_2}(\mathcal{H})\big) \le \gamma(h)\big(\mathcal{E}_{\ell_1}(h) - \mathcal{E}_{\ell_1}^*(\mathcal{H}) + \mathcal{M}_{\ell_1}(\mathcal{H})\big). \tag{3}$$

*with* $\gamma(h) = \big[\sup_{x\in\mathcal{X}}\alpha(h,x)\beta(h,x)\big]$. *If, additionally,* $\mathcal{X}$ *is a subset of* $\mathbb{R}^n$ *and, for any* $h\in\mathcal{H}$, $x\mapsto\Delta\mathcal{C}_{\ell_1,\mathcal{H}}(h,x)$ *is non-decreasing and* $x\mapsto\alpha(h,x)\beta(h,x)$ *is non-increasing, or vice-versa, then, the inequality holds with* $\gamma(h) = \mathbb{E}_X[\alpha(h,x)\beta(h,x)]$.

**Proof** For any $h\in\mathcal{H}$, we can write

$$
\begin{aligned}
\Psi\big(\mathcal{E}_{\ell_2}(h) - \mathcal{E}_{\ell_2}^*(\mathcal{H}) + \mathcal{M}_{\ell_2}(\mathcal{H})\big) &= \Psi\left(\mathbb{E}_X\big[\Delta\mathcal{C}_{\ell_2,\mathcal{H}}(h,x)\big]\right) \\
&= \Psi\left(\mathbb{E}_X\left[\beta(h,x)\frac{\Delta\mathcal{C}_{\ell_2,\mathcal{H}}(h,x)}{\beta(h,x)}\right]\right) \\
&\le \mathbb{E}_X\left[\beta(h,x)\Psi\left(\frac{\Delta\mathcal{C}_{\ell_2,\mathcal{H}}(h,x)}{\beta(h,x)}\right)\right] \qquad \text{(Jensen's ineq.)} \\
&\le \mathbb{E}_X\big[\alpha(h,x)\beta(h,x)\Delta\mathcal{C}_{\ell_1,\mathcal{H}}(h,x)\big] \qquad \text{(assumption)} \\
&\le \left[\sup_{x\in\mathcal{X}}\alpha(h,x)\beta(h,x)\right]\mathbb{E}_X\big[\Delta\mathcal{C}_{\ell_1,\mathcal{H}}(h,x)\big] \quad \text{(Hölder's ineq.)} \\
&= \left[\sup_{x\in\mathcal{X}}\alpha(h,x)\beta(h,x)\right]\big(\mathcal{E}_{\ell_1}(h) - \mathcal{E}_{\ell_1}^*(\mathcal{H}) + \mathcal{M}_{\ell_1}(\mathcal{H})\big). \\
&\hspace{5cm} \text{(def. of } \mathbb{E}_X\big[\Delta\mathcal{C}_{\ell_1,\mathcal{H}}(h,x)\big])
\end{aligned}
$$

If, additionally, $\mathcal{X}$ is a subset of $\mathbb{R}^n$ and, for any $h\in\mathcal{H}$, $x\mapsto\Delta\mathcal{C}_{\ell_1,\mathcal{H}}(h,x)$ is non-decreasing and $x\mapsto\alpha(h,x)\beta(h,x)$ is non-increasing, or vice-versa, then, by the FKG inequality (Fortuin et al., 1971), we have

$$
\begin{aligned}
\Psi\big(\mathcal{E}_{\ell_2}(h) - \mathcal{E}_{\ell_2}^*(\mathcal{H}) + \mathcal{M}_{\ell_2}(\mathcal{H})\big) &\le \mathbb{E}_X\big[\alpha(h,x)\beta(h,x)\Delta\mathcal{C}_{\ell_1,\mathcal{H}}(h,x)\big] \\
&\le \mathbb{E}_X[\alpha(h,x)\beta(h,x)]\,\mathbb{E}_X\big[\Delta\mathcal{C}_{\ell_1,\mathcal{H}}(h,x)\big] \\
&\le \mathbb{E}_X[\alpha(h,x)\beta(h,x)]\big(\mathcal{E}_{\ell_1}(h) - \mathcal{E}_{\ell_1}^*(\mathcal{H}) + \mathcal{M}_{\ell_1}(\mathcal{H})\big),
\end{aligned}
$$

which completes the proof. $\blacksquare$

**Theorem 2** *Assume that there exist a concave function* $\Gamma\colon\mathbb{R}_+ \to \mathbb{R}$ *and two positive functions* $\alpha\colon\mathcal{H}\times\mathcal{X}\to\mathbb{R}_+^*$ *and* $\beta\colon\mathcal{H}\times\mathcal{X}\to\mathbb{R}_+^*$ *with* $\sup_{x\in\mathcal{X}}\alpha(h,x)<+\infty$ *and* $\mathbb{E}_{x\in\mathcal{X}}[\beta(h,x)]=1$ *for all* $h\in\mathcal{H}$ *such*

*that the following holds for all $h \in \mathcal{H}$ and $x \in \mathcal{X}$: $\frac{\Delta \mathcal{C}_{\ell_2,\mathcal{H}}(h,x)}{\beta(h,x)} \leq \Gamma\big(\alpha(h,x)\,\Delta \mathcal{C}_{\ell_1,\mathcal{H}}(h,x)\big)$. Then, the following inequality holds for any hypothesis $h \in \mathcal{H}$:*

$$\mathcal{E}_{\ell_2}(h) - \mathcal{E}_{\ell_2}^*(\mathcal{H}) + \mathcal{M}_{\ell_2}(\mathcal{H}) \leq \Gamma\big(\gamma(h)\big(\mathcal{E}_{\ell_1}(h) - \mathcal{E}_{\ell_1}^*(\mathcal{H}) + \mathcal{M}_{\ell_1}(\mathcal{H})\big)\big), \tag{4}$$

*with $\gamma(h) = [\sup_{x \in \mathcal{X}} \alpha(h,x)\beta(h,x)]$. If, additionally, $\mathcal{X}$ is a subset of $\mathbb{R}^n$ and, for any $h \in \mathcal{H}$, $x \mapsto \Delta \mathcal{C}_{\ell_1,\mathcal{H}}(h,x)$ is non-decreasing and $x \mapsto \alpha(h,x)\beta(h,x)$ is non-increasing, or vice-versa, then, the inequality holds with $\gamma(h) = \mathbb{E}_X[\alpha(h,x)\beta(h,x)]$.*

**Proof** For any $h \in \mathcal{H}$, we can write

$$\begin{aligned}
\mathcal{E}_{\ell_2}(h) - \mathcal{E}_{\ell_2}^*(\mathcal{H}) + \mathcal{M}_{\ell_2}(\mathcal{H}) &= \mathbb{E}_X\big[\Delta \mathcal{C}_{\ell_2,\mathcal{H}}(h,x)\big] \\
&\leq \mathbb{E}_X\big[\beta(h,x)\Gamma\big(\alpha(h,x)\,\Delta \mathcal{C}_{\ell_1,\mathcal{H}}(h,x)\big)\big] && \text{(assumption)} \\
&\leq \Gamma\bigg(\mathbb{E}_X\big[\alpha(h,x)\beta(h,x)\Delta \mathcal{C}_{\ell_1,\mathcal{H}}(h,x)\big]\bigg) \\
&&& \text{(Jensen's ineq.)} \\
&\leq \Gamma\bigg(\Big[\sup_{x \in \mathcal{X}}\alpha(h,x)\beta(h,x)\Big]\mathbb{E}_X\big[\Delta \mathcal{C}_{\ell_1,\mathcal{H}}(h,x)\big]\bigg) && \text{(Hölder's ineq.)} \\
&= \Gamma\bigg(\Big[\sup_{x \in \mathcal{X}}\alpha(h,x)\beta(h,x)\Big]\big(\mathcal{E}_{\ell_1}(h) - \mathcal{E}_{\ell_1}^*(\mathcal{H}) + \mathcal{M}_{\ell_1}(\mathcal{H})\big)\bigg). \\
&&& (\text{def. of } \mathbb{E}_X\big[\Delta \mathcal{C}_{\ell_1,\mathcal{H}}(h,x)\big])
\end{aligned}$$

If, additionally, $\mathcal{X}$ is a subset of $\mathbb{R}^n$ and, for any $h \in \mathcal{H}$, $x \mapsto \Delta \mathcal{C}_{\ell_1,\mathcal{H}}(h,x)$ is non-decreasing and $x \mapsto \alpha(h,x)\beta(h,x)$ is non-increasing, or vice-versa, then, by the FKG inequality (Fortuin et al., 1971), we have

$$\begin{aligned}
\mathcal{E}_{\ell_2}(h) - \mathcal{E}_{\ell_2}^*(\mathcal{H}) + \mathcal{M}_{\ell_2}(\mathcal{H}) &\leq \Gamma\bigg(\mathbb{E}_X\big[\alpha(h,x)\beta(h,x)\Delta \mathcal{C}_{\ell_1,\mathcal{H}}(h,x)\big]\bigg) \\
&\leq \Gamma\bigg(\mathbb{E}_X\big[\alpha(h,x)\beta(h,x)\big]\mathbb{E}_X\big[\Delta \mathcal{C}_{\ell_1,\mathcal{H}}(h,x)\big]\bigg) \\
&\leq \Gamma\bigg(\mathbb{E}_X\big[\alpha(h,x)\beta(h,x)\big]\big(\mathcal{E}_{\ell_1}(h) - \mathcal{E}_{\ell_1}^*(\mathcal{H}) + \mathcal{M}_{\ell_1}(\mathcal{H})\big)\bigg),
\end{aligned}$$

which completes the proof. ∎

**Lemma 3** *The bounds of Theorems 1 and 2 are* tight *in the following sense: for some distributions, Inequality (3) (respectively Inequality (4)) is the tightest possible $\mathcal{H}$-consistency bound that can be derived under the assumption of Theorem 1 (respectively Theorem 2).*

**Proof** Take $h$ and $x$ such that $\sup_{x \in \mathcal{X}} \alpha(h,x)\beta(h,x)$ is achieved. Consider the distribution concentrates on that $x$. Then, the bounds given by (3) and (4) reduce to the following forms:

$$\Psi\bigg(\frac{\Delta \mathcal{C}_{\ell_2,\mathcal{H}}(h,x)}{\beta(h,x)}\bigg) \leq \alpha(h,x)\,\Delta \mathcal{C}_{\ell_1,\mathcal{H}}(h,x)$$

$$\frac{\Delta \mathcal{C}_{\ell_2,\mathcal{H}}(h,x)}{\beta(h,x)} \leq \Gamma\big(\alpha(h,x)\,\Delta \mathcal{C}_{\ell_1,\mathcal{H}}(h,x)\big)$$

where we used the fact that $\mathbb{E}_X[\beta(h,x)] = 1$ in this case. They exactly match the assumptions in Theorems 1 and 2, which are the tightest inequalities that can be obtained. The $\mathcal{H}$-consistency bound is in fact an equality in the same cases when the assumption holds with the best choices of $\alpha$ and $\beta$. ∎

**Theorem 4** *Assume that there exist two positive functions $\alpha\colon \mathcal{H} \times \mathcal{X} \to \mathbb{R}_+^*$ and $\beta\colon \mathcal{H} \times \mathcal{X} \to \mathbb{R}_+^*$ with $\sup_{x \in \mathcal{X}} \alpha(h,x) < +\infty$ and $\mathbb{E}_{x \in \mathcal{X}}[\beta(h,x)] = 1$ for all $h \in \mathcal{H}$ such that the following holds for all $h \in \mathcal{H}$ and $x \in \mathcal{X}$: $\frac{\Delta\mathcal{C}_{\ell_2,\mathcal{H}}(h,x)}{\beta(h,x)} \leq \left(\alpha(h,x)\,\Delta\mathcal{C}_{\ell_1,\mathcal{H}}(h,x)\right)^{\frac{1}{s}}$, for some $s \geq 1$ with conjugate number $t \geq 1$, that is $\frac{1}{s} + \frac{1}{t} = 1$. Then, for $\gamma(h) = \mathbb{E}_X\left[\alpha^{\frac{t}{s}}(h,x)\beta^t(h,x)\right]^{\frac{1}{t}}$, the following inequality holds for any $h \in \mathcal{H}$:*

$$\mathcal{E}_{\ell_2}(h) - \mathcal{E}_{\ell_2}^*(\mathcal{H}) + \mathcal{M}_{\ell_2}(\mathcal{H}) \leq \gamma(h)\left[\mathcal{E}_{\ell_1}(h) - \mathcal{E}_{\ell_1}^*(\mathcal{H}) + \mathcal{M}_{\ell_1}(\mathcal{H})\right]^{\frac{1}{s}}.$$

**Proof** For any $h \in \mathcal{H}$, we can write

$$\begin{aligned}
\mathcal{E}_{\ell_2}(h) - \mathcal{E}_{\ell_2}^*(\mathcal{H}) + \mathcal{M}_{\ell_2}(\mathcal{H}) &= \mathbb{E}_X\left[\Delta\mathcal{C}_{\ell_2,\mathcal{H}}(h,x)\right] \\
&\leq \mathbb{E}_X\left[\beta(h,x)\alpha^{\frac{1}{s}}(h,x)\,\Delta\mathcal{C}_{\ell_1,\mathcal{H}}^{\frac{1}{s}}(h,x)\right] && \text{(assumption)} \\
&\leq \mathbb{E}_X\left[\alpha^{\frac{t}{s}}(h,x)\beta^t(h,x)\right]^{\frac{1}{t}} \mathbb{E}_X\left[\Delta\mathcal{C}_{\ell_1,\mathcal{H}}(h,x)\right]^{\frac{1}{s}} && \text{(Hölder's ineq.)} \\
&= \mathbb{E}_X\left[\alpha^{\frac{t}{s}}(h,x)\beta^t(h,x)\right]^{\frac{1}{t}} \left(\mathcal{E}_{\ell_1}(h) - \mathcal{E}_{\ell_1}^*(\mathcal{H}) + \mathcal{M}_{\ell_1}(\mathcal{H})\right)^{\frac{1}{s}}. \\
&&& \hspace{-3cm}\text{(def. of } \mathbb{E}_X\left[\Delta\mathcal{C}_{\ell_1,\mathcal{H}}(h,x)\right])
\end{aligned}$$

This completes the proof. ∎

## Appendix C. Proof of enhanced $\mathcal{H}$-consistency bounds in multi-class classification (Theorem 5)

To begin the proof, we first introduce the following result from Awasthi et al. (2022b), which characterizes the conditional regret of the multi-class zero-one loss. For completeness, we include the proof here.

**Lemma 16** *Assume that $\mathcal{H}$ is symmetric and complete. For any $x \in \mathcal{X}$, the best-in-class conditional error and the conditional regret for $\ell_{0-1}$ can be expressed as follows:*

$$\mathcal{C}_{\ell_{0-1},\mathcal{H}}^*(x) = 1 - \max_{y \in \mathcal{Y}} p(y \mid x)$$

$$\Delta\mathcal{C}_{\ell_{0-1},\mathcal{H}}(h,x) = \max_{y \in \mathcal{Y}} p(y \mid x) - p(\mathsf{h}(x) \mid x)$$

**Proof** The conditional error for $\ell_{0-1}$ can be expressed as:

$$\mathcal{C}_{\ell_{0-1}}(h,x) = \sum_{y \in \mathcal{Y}} p(y \mid x) 1_{\mathsf{h}(x) \neq y} = 1 - p(\mathsf{h}(x) \mid y).$$

Since $\mathcal{H}$ is symmetric and complete, we have $\{h(x): h \in \mathcal{H}\} = \mathcal{Y}$. Therefore,

$$\mathcal{C}^*_{\ell_{0-1},\mathcal{H}}(x) = \inf_{h \in \mathcal{H}} \mathcal{C}_{\ell_{0-1}}(h, x) = 1 - \max_{y \in \mathcal{Y}} p(y \mid x)$$

$$\Delta\mathcal{C}_{\ell_{0-1},\mathcal{H}}(h, x) = \mathcal{C}_{\ell_{0-1}}(h, x) - \mathcal{C}^*_{\ell_{0-1},\mathcal{H}}(x) = \max_{y \in \mathcal{Y}} p(y \mid x) - p(h(x) \mid x).$$

This completes the proof. ∎

**Theorem 5 (Enhanced $\mathcal{H}$-consistency bounds for constrained losses)** *Assume that $\mathcal{H}$ is symmetric and complete. Then, the following inequality holds for any hypothesis $h \in \mathcal{H}$:*

$$\mathcal{E}_{\ell_{0-1}}(h) - \mathcal{E}^*_{\ell_{0-1}}(\mathcal{H}) + \mathcal{M}_{\ell_{0-1}}(\mathcal{H}) \leq \Gamma\big(\mathcal{E}_{\Phi\text{cstnd}}(h) - \mathcal{E}^*_{\Phi\text{cstnd}}(\mathcal{H}) + \mathcal{M}_{\Phi\text{cstnd}}(\mathcal{H})\big),$$

*where $\Gamma(x) = \dfrac{\sqrt{2}\,x^{\frac{1}{2}}}{\left(e^{\Lambda(h)}\right)^{\frac{1}{2}}}$ for $\Phi(u) = e^{-u}$, $\Gamma(x) = \dfrac{x}{1+\Lambda(h)}$ for $\Phi(u) = \max\{0, 1 - u\}$, and $\Gamma(x) = \dfrac{x^{\frac{1}{2}}}{1+\Lambda(h)}$ for $\Phi(u) = (1 - u)^2 1_{u \leq 1}$. Additionally, $\Lambda(h) = \inf_{x \in \mathcal{X}} \max_{y \in \mathcal{Y}} h(x, y)$.*

**Proof** The conditional error for constrained losses can be expressed as follows:

$$\mathcal{C}_{\Phi\text{cstnd}}(h, x) = \sum_{y \in \mathcal{Y}} p(y \mid x) \sum_{y' \neq y} \Phi\big(-h(x, y')\big) = \sum_{y \in \mathcal{Y}} (1 - p(y \mid x))\Phi(-h(x, y)).$$

Next, we will provide the proof for each case individually. We denote by $y_{\max} = \text{argmax}_{y \in \mathcal{Y}} p(y \mid x)$.

**Constrained exponential loss with $\Phi(u) = e^{-u}$.** When $h(x) = y_{\max}$, we have $\Delta\mathcal{C}_{\ell_{0-1},\mathcal{H}}(h, x) = 0$. Let $h \in \mathcal{H}$ be a hypothesis such that $h(x) \neq y_{\max}$. In this case, the conditional error can be written as:

$$\mathcal{C}_{\Phi\text{cstnd}}(h, x) = \sum_{y \in \mathcal{Y}} (1 - p(y \mid x))e^{h(x,y)} = \sum_{y \in \{y_{\max}, h(x)\}} (1 - p(y \mid x))e^{h(x,y)} + \sum_{y \notin \{y_{\max}, h(x)\}} e^{h(x,y)}.$$

For any $x \in \mathcal{X}$, define the hypothesis $h_\mu \in \mathcal{H}$ by

$$h_\mu(x, y) = \begin{cases} h(x, y) & \text{if } y \notin \{y_{\max}, h(x)\} \\ h(x, y_{\max}) + \mu & \text{if } y = h(x) \\ h(x, h(x)) - \mu & \text{if } y = y_{\max} \end{cases}$$

for any $\mu \in \mathbb{R}$. By the completeness of $\mathcal{H}$, we have $h_\mu \in \mathcal{H}$ and $\sum_{y \in \mathcal{Y}} h_\mu(x, y) = 0$. Thus,

$$
\begin{aligned}
\Delta \mathcal{C}_{\Phi^{\text{cstnd}}, \mathcal{H}}(h, x) &= \mathcal{C}_{\Phi^{\text{cstnd}}}(h, x) - \mathcal{C}_{\Phi^{\text{cstnd}}}^*(\mathcal{H}, x) \\
&\geq \mathcal{C}_{\Phi^{\text{cstnd}}}(h, x) - \inf_{\mu \in \mathbb{R}} \mathcal{C}_{\Phi^{\text{cstnd}}}(h_\mu, x) \\
&= \left( \sqrt{(1 - p(\mathsf{h}(x) \mid x)) e^{h(x, \mathsf{h}(x))}} - \sqrt{(1 - p(y_{\max} \mid x)) e^{h(x, y_{\max})}} \right)^2 \\
&\geq e^{h(x, \mathsf{h}(x))} \left( \sqrt{(1 - p(\mathsf{h}(x) \mid x))} - \sqrt{(1 - p(y_{\max} \mid x))} \right)^2 \\
&\qquad\qquad (e^{h(x, \mathsf{h}(x))} \geq e^{h(x, y_{\max})} \text{ and } p(\mathsf{h}(x) \mid x) \leq p(y_{\max} \mid x)) \\
&= e^{h(x, \mathsf{h}(x))} \left( \frac{p(y_{\max} \mid x) - p(\mathsf{h}(x) \mid x)}{\sqrt{(1 - p(\mathsf{h}(x) \mid x))} + \sqrt{(1 - p(y_{\max} \mid x))}} \right)^2 \\
&\geq \frac{e^{h(x, \mathsf{h}(x))}}{2} \left( \max_{y \in \mathcal{Y}} p(x, y) - p(\mathsf{h}(x) \mid x) \right)^2 \quad (0 \leq p(y_{\max} \mid x) + p(\mathsf{h}(x) \mid x) \leq 1) \\
&\geq \frac{e^{\Lambda(h)}}{2} \left( \Delta \mathcal{C}_{\ell_{0-1}, \mathcal{H}}(h, x) \right)^2.
\end{aligned}
$$

Therefore, by Theorems 1 or 2, the following inequality holds for any hypothesis $h \in \mathcal{H}$:

$$
\mathcal{E}_{\ell_{0-1}}(h) - \mathcal{E}_{\ell_{0-1}}^*(\mathcal{H}) + \mathcal{M}_{\ell_{0-1}}(\mathcal{H}) \leq \frac{\sqrt{2}}{\left( e^{\Lambda(h)} \right)^{\frac{1}{2}}} \left( \mathcal{E}_{\Phi^{\text{cstnd}}}(h) - \mathcal{E}_{\Phi^{\text{cstnd}}}^*(\mathcal{H}) + \mathcal{M}_{\Phi^{\text{cstnd}}}(\mathcal{H}) \right)^{\frac{1}{2}}.
$$

**Constrained hinge loss with $\Phi(u) = \max\{1 - u, 0\}$.** When $\mathsf{h}(x) = y_{\max}$, we have $\Delta \mathcal{C}_{\ell_{0-1}, \mathcal{H}}(h, x) = 0$. Let $h \in \mathcal{H}$ be a hypothesis such that $\mathsf{h}(x) \neq y_{\max}$. In this case, the conditional error can be written as:

$$
\begin{aligned}
\mathcal{C}_{\Phi^{\text{cstnd}}}(h, x) &= \sum_{y \in \mathcal{Y}} (1 - p(y \mid x)) \max\{1 + h(x, y), 0\} \\
&= \sum_{y \in \{y_{\max}, \mathsf{h}(x)\}} (1 - p(y \mid x)) \max\{1 + h(x, y), 0\} + \sum_{y \notin \{y_{\max}, \mathsf{h}(x)\}} \max\{1 + h(x, y), 0\}.
\end{aligned}
$$

For any $x \in \mathcal{X}$, define the hypothesis $h_\mu \in \mathcal{H}$ by

$$
h_\mu(x, y) = \begin{cases} h(x, y) & \text{if } y \notin \{y_{\max}, \mathsf{h}(x)\} \\ h(x, y_{\max}) + \mu & \text{if } y = \mathsf{h}(x) \\ h(x, \mathsf{h}(x)) - \mu & \text{if } y = y_{\max} \end{cases}
$$

for any $\mu \in \mathbb{R}$. By the completeness of $\mathcal{H}$, we have $h_\mu \in \mathcal{H}$ and $\sum_{y \in \mathcal{Y}} h_\mu(x, y) = 0$. Thus,

$$
\begin{aligned}
\Delta \mathcal{C}_{\Phi^{\text{cstnd}}, \mathcal{H}}(h, x) &= \mathcal{C}_{\Phi^{\text{cstnd}}}(h, x) - \mathcal{C}_{\Phi^{\text{cstnd}}}^*(\mathcal{H}, x) \\
&\geq \mathcal{C}_{\Phi^{\text{cstnd}}}(h, x) - \inf_{\mu \in \mathbb{R}} \mathcal{C}_{\Phi^{\text{cstnd}}}(h_\mu, x) \\
&\geq (1 + h(x, \mathsf{h}(x)))(p(y_{\max} \mid x) - p(\mathsf{h}(x) \mid x)) \\
&\geq (1 + \Lambda(h)) \left( \Delta \mathcal{C}_{\ell_{0-1}, \mathcal{H}}(h, x) \right).
\end{aligned}
$$

Therefore, by Theorems 1 or 2, the following inequality holds for any hypothesis $h \in \mathcal{H}$:

$$\mathcal{E}_{\ell_{0-1}}(h) - \mathcal{E}_{\ell_{0-1}}^*(\mathcal{H}) + \mathcal{M}_{\ell_{0-1}}(\mathcal{H}) \le \frac{1}{1 + \Lambda(h)} \big( \mathcal{E}_{\Phi^{\mathrm{cstnd}}}(h) - \mathcal{E}_{\Phi^{\mathrm{cstnd}}}^*(\mathcal{H}) + \mathcal{M}_{\Phi^{\mathrm{cstnd}}}(\mathcal{H}) \big).$$

**Constrained squared hinge loss with** $\Phi(u) = (1 - u)^2 1_{u \le 1}$**.** When $\mathsf{h}(x) = y_{\max}$, we have $\Delta \mathcal{C}_{\ell_{0-1}, \mathcal{H}}(h, x) = 0$. Let $h \in \mathcal{H}$ be a hypothesis such that $\mathsf{h}(x) \ne y_{\max}$. In this case, the conditional error can be written as:

$$\begin{aligned}
\mathcal{C}_{\Phi^{\mathrm{cstnd}}}(h, x) &= \sum_{y \in \mathcal{Y}} (1 - p(y \mid x)) \max\{1 + h(x, y), 0\}^2 \\
&= \sum_{y \in \{y_{\max}, \mathsf{h}(x)\}} (1 - p(y \mid x)) \max\{1 + h(x, y), 0\}^2 + \sum_{y \notin \{y_{\max}, \mathsf{h}(x)\}} \max\{1 + h(x, y), 0\}^2.
\end{aligned}$$

For any $x \in \mathcal{X}$, define the hypothesis $h_\mu \in \mathcal{H}$ by

$$h_\mu(x, y) = \begin{cases} h(x, y) & \text{if } y \notin \{y_{\max}, \mathsf{h}(x)\} \\ h(x, y_{\max}) + \mu & \text{if } y = \mathsf{h}(x) \\ h(x, \mathsf{h}(x)) - \mu & \text{if } y = y_{\max} \end{cases}$$

for any $\mu \in \mathbb{R}$. By the completeness of $\mathcal{H}$, we have $h_\mu \in \mathcal{H}$ and $\sum_{y \in \mathcal{Y}} h_\mu(x, y) = 0$. Thus,

$$\begin{aligned}
\Delta \mathcal{C}_{\Phi^{\mathrm{cstnd}}, \mathcal{H}}(h, x) &= \mathcal{C}_{\Phi^{\mathrm{cstnd}}}(h, x) - \mathcal{C}_{\Phi^{\mathrm{cstnd}}}^*(\mathcal{H}, x) \\
&\ge \mathcal{C}_{\Phi^{\mathrm{cstnd}}}(h, x) - \inf_{\mu \in \mathbb{R}} \mathcal{C}_{\Phi^{\mathrm{cstnd}}}(h_\mu, x) \\
&\ge (1 + h(x, \mathsf{h}(x)))^2 (p(y_{\max} \mid x) - p(\mathsf{h}(x) \mid x))^2 \\
&\ge (1 + \Lambda(h))^2 \big( \Delta \mathcal{C}_{\ell_{0-1}, \mathcal{H}}(h, x) \big)^2.
\end{aligned}$$

Therefore, by Theorems 1 or 2, the following inequality holds for any hypothesis $h \in \mathcal{H}$:

$$\mathcal{E}_{\ell_{0-1}}(h) - \mathcal{E}_{\ell_{0-1}}^*(\mathcal{H}) + \mathcal{M}_{\ell_{0-1}}(\mathcal{H}) \le \frac{1}{1 + \Lambda(h)} \big( \mathcal{E}_{\Phi^{\mathrm{cstnd}}}(h) - \mathcal{E}_{\Phi^{\mathrm{cstnd}}}^*(\mathcal{H}) + \mathcal{M}_{\Phi^{\mathrm{cstnd}}}(\mathcal{H}) \big)^{\frac{1}{2}}.$$

$\blacksquare$

# Appendix D. Proof of enhanced $\mathcal{H}$-consistency bounds under low-noise conditions

## D.1. Proof of Theorem 6

**Theorem 6** *Consider a binary classification setting where the Tsybakov noise assumption holds. Assume that the following holds for all $h \in \mathcal{H}$ and $x \in \mathcal{X}$: $\Delta \mathcal{C}_{\ell_{0-1}^{\mathrm{bi}}, \mathcal{H}}(h, x) \le \Gamma \big( \Delta \mathcal{C}_{\ell, \mathcal{H}}(h, x) \big)$, with $\Gamma(x) = x^{\frac{1}{s}}$, for some $s \ge 1$. Then, for any $h \in \mathcal{H}$,*

$$\mathcal{E}_{\ell_{0-1}^{\mathrm{bi}}}(h) - \mathcal{E}_{\ell_{0-1}^{\mathrm{bi}}}^*(\mathcal{H}) \le c^{\frac{s-1}{s - \alpha(s-1)}} \big[ \mathcal{E}_\ell(h) - \mathcal{E}_\ell^*(\mathcal{H}) + \mathcal{M}_\ell(\mathcal{H}) \big]^{\frac{1}{s - \alpha(s-1)}}.$$

**Proof** Fix $\epsilon > 0$ and define $\beta(h,x) = \frac{1_{h(x)\neq h^*(x)}+\epsilon}{\mathbb{E}_X[1_{h(x)\neq h^*(x)}+\epsilon]}$. Since $\Delta\mathcal{C}_{\ell^{bi}_{0-1},\mathcal{H}}(h,x) = |2\eta(x)-1|1_{h(x)\neq h^*(x)}$,

we have $\frac{\Delta\mathcal{C}_{\ell^{bi}_{0-1},\mathcal{H}}(h,x)}{\beta(h,x)} \leq \Delta\mathcal{C}_{\ell^{bi}_{0-1},\mathcal{H}}(h,x)\,\mathbb{E}_X[1_{h(x)\neq h^*(x)}+\epsilon]$, thus the following inequality holds

$$\frac{\Delta\mathcal{C}_{\ell^{bi}_{0-1},\mathcal{H}}(h,x)}{\beta(h,x)} \leq \mathbb{E}_X[1_{h(x)\neq h^*(x)}+\epsilon]\Delta\mathcal{C}^{\frac{1}{s}}_{\ell,\mathcal{H}}(h,x).$$

By Theorem 4, with $\alpha(h,x) = \mathbb{E}_X[1_{h(x)\neq h^*(x)}+\epsilon]^s$, we have

$$\mathcal{E}_{\ell^{bi}_{0-1}}(h) - \mathcal{E}^*_{\ell^{bi}_{0-1}}(\mathcal{H}) \leq \left(\mathbb{E}_X[1_{h(x)\neq h^*(x)}+\epsilon]^t\right)^{\frac{1}{t}}(\mathcal{E}_\ell(h) - \mathcal{E}^*_\ell(\mathcal{H}) + \mathcal{M}_\ell(\mathcal{H}))^{\frac{1}{s}}.$$

Since the inequality holds for any $\epsilon > 0$, it implies:

$$\mathcal{E}_{\ell^{bi}_{0-1}}(h) - \mathcal{E}^*_{\ell^{bi}_{0-1}}(\mathcal{H}) \leq \mathbb{E}_X\left[\left(1_{h(x)\neq h^*(x)}\right)^t\right]^{\frac{1}{t}}(\mathcal{E}_\ell(h) - \mathcal{E}^*_\ell(\mathcal{H}) + \mathcal{M}_\ell(\mathcal{H}))^{\frac{1}{s}}$$

$$= \mathbb{E}_X[1_{h(x)\neq h^*(x)}]^{\frac{1}{t}}(\mathcal{E}_\ell(h) - \mathcal{E}^*_\ell(\mathcal{H}) + \mathcal{M}_\ell(\mathcal{H}))^{\frac{1}{s}}$$

$$\left(\left(1_{h(x)\neq h^*(x)}\right)^t = 1_{h(x)\neq h^*(x)}\right)$$

$$\leq c^{\frac{1}{t}}\left[\mathcal{E}_{\ell^{bi}_{0-1}}(h) - \mathcal{E}^*_{\ell^{bi}_{0-1}}(\mathcal{H})\right]^{\frac{\alpha}{t}}(\mathcal{E}_\ell(h) - \mathcal{E}^*_\ell(\mathcal{H}) + \mathcal{M}_\ell(\mathcal{H}))^{\frac{1}{s}}$$

(Tsybakov noise assumption)

The result follows after dividing both sides by $\left[\mathcal{E}_{\ell^{bi}_{0-1}}(h) - \mathcal{E}^*_{\ell^{bi}_{0-1}}(\mathcal{H})\right]^{\frac{\alpha}{t}}$. ∎

### D.2. Proof of Lemma 7 and Lemma 8

**Lemma 7** *The Tsybakov noise assumption implies that there exists a constant $c$ such that the following inequalities hold for any $h \in \mathcal{H}$:*

$$\mathbb{E}[1_{h(x)\neq h^*(x)}] \leq c\,\mathbb{E}[\gamma(X)1_{h(x)\neq h^*(x)}]^\alpha \leq c[\mathcal{E}_{\ell_{0-1}}(h) - \mathcal{E}_{\ell_{0-1}}(h^*)]^\alpha.$$

**Proof** We prove the first inequality, the second one follows immediately the definition of the margin. By definition of the expectation and the Lebesgue integral, for any $u > 0$, we can write

$$\mathbb{E}[\gamma(X)1_{h(X)\neq h^*(X)}] = \int_0^{+\infty} \mathbb{P}[\gamma(X)1_{h(X)\neq h^*(X)} > t]\,dt$$

$$\geq \int_0^u \mathbb{P}[\gamma(X)1_{h(X)\neq h^*(X)} > t]\,dt$$

$$= \int_0^u \mathbb{E}[1_{\gamma(X)>t}\,1_{h(X)\neq h^*(X)}]\,dt$$

$$= \int_0^u \left(\mathbb{E}[1_{\gamma(X)>t}] - \mathbb{E}[1_{\gamma(X)>t}1_{h(X)=h^*(X)}]\right)dt$$

$$\geq \int_0^u \left(\mathbb{E}[1_{\gamma(X)>t}] - \mathbb{E}[1_{h(X)=h^*(X)}]\right)dt$$

$$= \int_0^u \left(\mathbb{P}[h(X)\neq h^*(X)] - \mathbb{P}[\gamma(X)\leq t]\right)dt$$

$$\geq u\,\mathbb{P}[h(X)\neq h^*(X)] - \int_0^u Bt^{\frac{\alpha}{1-\alpha}}\,dt$$

$$= u\,\mathbb{P}[h(X)\neq h^*(X)] - B(1-\alpha)u^{\frac{1}{1-\alpha}}.$$

Taking the derivative and choosing $u$ to maximize the above gives $u = \left[\frac{\mathbb{P}[\mathsf{h}(X) \neq \mathsf{h}^*(X)]}{B}\right]^{\frac{1-\alpha}{\alpha}}$. Plugging in this choice of $u$ gives

$$\mathbb{E}[\gamma(X)1_{\mathsf{h}(X) \neq \mathsf{h}^*(X)}] \geq \left[\frac{1}{B}\right]^{\frac{1-\alpha}{\alpha}} \alpha\, \mathbb{P}[\mathsf{h}(X) \neq \mathsf{h}^*(X)]^{\frac{1}{\alpha}},$$

which can be rewritten as $\mathbb{P}[\mathsf{h}(X) \neq \mathsf{h}^*(X)] \leq c\, \mathbb{E}[\gamma(X)1_{\mathsf{h}(X) \neq \mathsf{h}^*(X)}]^{\alpha}$ for $c = \frac{B^{1-\alpha}}{\alpha^\alpha}$. ∎

**Lemma 8** *Assume that for any $h \in \mathcal{H}_{\text{all}}$, we have $\mathbb{P}[\mathsf{h}(X) \neq \mathsf{h}^*(X)] \leq c\, \mathbb{E}[\gamma(X)1_{\gamma(X) \leq t}]^{\alpha}$. Then, the Tsybakov noise condition holds, that is, there exists a constant $B > 0$, such that*

$$\forall t > 0, \quad \mathbb{P}[\gamma(X) \leq t] \leq Bt^{\frac{\alpha}{1-\alpha}}.$$

**Proof** Fix $t > 0$ and consider the event $\{\gamma(X) \leq t\}$. Since $h$ can be chosen to be any measurable function, there exists $h$ such $1_{\gamma(X) \leq t} = 1_{\mathsf{h}(X) \neq \mathsf{h}^*(X)}$. In view of that, we can write

$$\mathbb{P}[\gamma(X) \leq t] = \mathbb{E}[1_{\gamma(X) \leq t}] \leq c\, \mathbb{E}[\gamma(X)1_{\gamma(X) \leq t}]^{\alpha} \leq ct^{\alpha}\, \mathbb{E}[1_{\gamma(X) \leq t}]^{\alpha}.$$

Comparing the left- and right-hand sides gives immediately

$$\mathbb{P}[\gamma(X) \leq t] \leq c^{\frac{1}{1-\alpha}}\, t^{\frac{\alpha}{1-\alpha}}.$$

Choosing $B = c^{\frac{1}{1-\alpha}}$ completes the proof. ∎

### D.3. Proof of Theorem 9

**Theorem 9** *Consider a multi-class classification setting where the Tsybakov noise assumption holds. Assume that the following holds for all $h \in \mathcal{H}$ and $x \in \mathcal{X}$: $\Delta\mathcal{C}_{\ell_{0-1},\mathcal{H}}(h, x) \leq \Gamma\big(\Delta\mathcal{C}_{\ell,\mathcal{H}}(h, x)\big)$, with $\Gamma(x) = x^{\frac{1}{s}}$, for some $s \geq 1$. Then, for any $h \in \mathcal{H}$,*

$$\mathcal{E}_{\ell_{0-1}}(h) - \mathcal{E}^*_{\ell_{0-1}}(\mathcal{H}) \leq c^{\frac{s-1}{s-\alpha(s-1)}}\big[\mathcal{E}_\ell(h) - \mathcal{E}^*_\ell(\mathcal{H}) + \mathcal{M}_\ell(\mathcal{H})\big]^{\frac{1}{s-\alpha(s-1)}}.$$

**Proof** Fix $\epsilon > 0$ and define $\beta(h, x) = \frac{1_{\mathsf{h}(x) \neq \mathsf{h}^*(x)} + \epsilon}{\mathbb{E}_X[1_{\mathsf{h}(X) \neq \mathsf{h}^*(X)} + \epsilon]}$. By Lemma 16, $\Delta\mathcal{C}_{\ell_{0-1},\mathcal{H}}(h, x) = \max_{y \in \mathcal{Y}} p(y \mid x) - p(\mathsf{h}(x) \mid x) = p(\mathsf{h}^*(x) \mid x) - p(\mathsf{h}(x) \mid x)$, we have

$$\frac{\Delta\mathcal{C}_{\ell_{0-1},\mathcal{H}}(h, x)}{\beta(h, x)} \leq \Delta\mathcal{C}_{\ell_{0-1},\mathcal{H}}(h, x)\, \mathbb{E}_X[1_{\mathsf{h}(X) \neq \mathsf{h}^*(X)} + \epsilon],$$

thus the following inequality holds

$$\frac{\Delta\mathcal{C}_{\ell_{0-1},\mathcal{H}}(h, x)}{\beta(h, x)} \leq \mathbb{E}_X[1_{\mathsf{h}(X) \neq \mathsf{h}^*(X)} + \epsilon]\Delta\mathcal{C}^{\frac{1}{s}}_{\ell,\mathcal{H}}(h, x).$$

By Theorem 4, with $\alpha(h, x) = \mathbb{E}_X[1_{\mathsf{h}(X) \neq \mathsf{h}^*(X)} + \epsilon]^s$, we have

$$\mathcal{E}_{\ell_{0-1}}(h) - \mathcal{E}^*_{\ell_{0-1}}(\mathcal{H}) \leq \left(\mathbb{E}_X[1_{\mathsf{h}(X) \neq \mathsf{h}^*(X)} + \epsilon]^t\right)^{\frac{1}{t}}(\mathcal{E}_\ell(h) - \mathcal{E}^*_\ell(\mathcal{H}) + \mathcal{M}_\ell(\mathcal{H}))^{\frac{1}{s}}.$$

Since the inequality holds for any $\epsilon > 0$, it implies:

$$
\begin{aligned}
\mathcal{E}_{\ell_{0-1}}(h) - \mathcal{E}_{\ell_{0-1}}^*(\mathcal{H}) &\leq \mathbb{E}_X\Big[\big(1_{\mathsf{h}(X)\neq\mathsf{h}^*(X)}\big)^t\Big]^{\frac{1}{t}} \big(\mathcal{E}_\ell(h) - \mathcal{E}_\ell^*(\mathcal{H}) + \mathcal{M}_\ell(\mathcal{H})\big)^{\frac{1}{s}} \\
&= \mathbb{E}_X\big[1_{\mathsf{h}(X)\neq\mathsf{h}^*(X)}\big]^{\frac{1}{t}} \big(\mathcal{E}_\ell(h) - \mathcal{E}_\ell^*(\mathcal{H}) + \mathcal{M}_\ell(\mathcal{H})\big)^{\frac{1}{s}} \\
&\qquad\qquad\qquad\qquad \Big(\big(1_{\mathsf{h}(X)\neq\mathsf{h}^*(X)}\big)^t = 1_{\mathsf{h}(x)\neq\mathsf{h}^*(x)}\Big) \\
&\leq c^{\frac{1}{t}}\big[\mathcal{E}_{\ell_{0-1}}(h) - \mathcal{E}_{\ell_{0-1}}^*(\mathcal{H})\big]^{\frac{\alpha}{t}} \big(\mathcal{E}_\ell(h) - \mathcal{E}_\ell^*(\mathcal{H}) + \mathcal{M}_\ell(\mathcal{H})\big)^{\frac{1}{s}} \\
&\qquad\qquad\qquad\qquad\qquad\qquad \text{(Tsybakov noise assumption)}
\end{aligned}
$$

The result follows after dividing both sides by $\big[\mathcal{E}_{\ell_{0-1}}(h) - \mathcal{E}_{\ell_{0-1}}^*(\mathcal{H})\big]^{\frac{\alpha}{t}}$. $\blacksquare$

## Appendix E. Proof of enhanced $\mathcal{H}$-consistency bounds in bipartite ranking

### E.1. Proof of Theorem 10

**Theorem 10** *Assume that there exist two concave functions $\Gamma_1\colon\mathbb{R}_+ \to \mathbb{R}$ and $\Gamma_2\colon\mathbb{R}_+ \to \mathbb{R}$, and two positive functions $\alpha_1\colon\mathcal{H} \times \mathcal{X} \to \mathbb{R}_+^*$ and $\alpha_2\colon\mathcal{H} \times \mathcal{X} \to \mathbb{R}_+^*$ with $\mathbb{E}_{x\in\mathcal{X}}[\alpha_1(h,x)] < +\infty$ and $\mathbb{E}_{x\in\mathcal{X}}[\alpha_2(h,x)] < +\infty$ for all $h \in \mathcal{H}$ such that the following holds for all $h \in \mathcal{H}$ and $(x,x') \in \mathcal{X} \times \mathcal{X}$: $\Delta\overline{\mathcal{C}}_{\mathsf{L},\mathcal{H}}(h,x,x') \leq \Gamma_1\big(\alpha_1(h,x')\,\Delta\mathcal{C}_{\ell,\mathcal{H}}(h,x)\big) + \Gamma_2\big(\alpha_2(h,x)\,\Delta\mathcal{C}_{\ell,\mathcal{H}}(h,x')\big)$. Then, the following inequality holds for any hypothesis $h \in \mathcal{H}$:*

$$
\mathcal{E}_{\mathsf{L}}(h) - \mathcal{E}_{\mathsf{L}}^*(\mathcal{H}) + \mathcal{M}_{\mathsf{L}}(\mathcal{H}) \leq \Gamma_1(\gamma_1(h)\mathsf{D}_\ell(h)) + \Gamma_2(\gamma_2(h)\mathsf{D}_\ell(h)).
$$

**Proof** For any $h \in \mathcal{H}$, we can write

$$
\begin{aligned}
&\mathcal{E}_{\mathsf{L}}(h) - \mathcal{E}_{\mathsf{L}}^*(\mathcal{H}) + \mathcal{M}_{\mathsf{L}}(\mathcal{H}) \\
&= \mathbb{E}_{X,X'}\big[\Delta\overline{\mathcal{C}}_{\mathsf{L},\mathcal{H}}(h,x,x')\big] \\
&\leq \mathbb{E}_{X,X'}\big[\Gamma_1\big(\alpha_1(h,x')\,\Delta\mathcal{C}_{\ell,\mathcal{H}}(h,x)\big) + \Gamma_2\big(\alpha_2(h,x)\,\Delta\mathcal{C}_{\ell,\mathcal{H}}(h,x')\big)\big] \qquad\text{(assumption)} \\
&\leq \Gamma_1\Big(\mathbb{E}_{X'}\big[\alpha_1(h,x')\big]\mathbb{E}_X\big[\Delta\mathcal{C}_{\ell,\mathcal{H}}(h,x)\big]\Big) + \Gamma_2\Big(\mathbb{E}_X\big[\alpha_2(h,x)\big]\mathbb{E}_{X'}\big[\Delta\mathcal{C}_{\ell,\mathcal{H}}(h,x')\big]\Big) \\
&\qquad\qquad\qquad\qquad\qquad\qquad\qquad\qquad\qquad\qquad\qquad\qquad \text{(Jensen's ineq.)} \\
&\leq \Gamma_1\Big(\mathbb{E}_{x\in\mathcal{X}}[\alpha_1(h,x)]\,(\mathcal{E}_\ell(h) - \mathcal{E}_\ell^*(\mathcal{H}) + \mathcal{M}_\ell(\mathcal{H}))\Big) \\
&\qquad + \Gamma_2\Big(\mathbb{E}_{x\in\mathcal{X}}[\alpha_2(h,x)]\,(\mathcal{E}_\ell(h) - \mathcal{E}_\ell^*(\mathcal{H}) + \mathcal{M}_\ell(\mathcal{H}))\Big), \qquad \text{(def. of } \mathbb{E}_X\big[\Delta\mathcal{C}_{\ell,\mathcal{H}}(h,x)\big])
\end{aligned}
$$

which completes the proof. $\blacksquare$

### E.2. Proof of Theorem 12

**Theorem 12** *Assume that $\mathcal{H}$ is complete. Then, the following inequality holds for the exponential loss $\Phi_{\exp}$:*

$$
\Delta\overline{\mathcal{C}}_{\mathsf{L}_{\Phi_{\exp}},\mathcal{H}}(h,x,x') \leq \mathcal{C}_{\ell_{\Phi_{\exp}}}(h,x')\,\Delta\mathcal{C}_{\ell_{\Phi_{\exp}},\mathcal{H}}(h,x) + \mathcal{C}_{\ell_{\Phi_{\exp}}}(h,x)\,\Delta\mathcal{C}_{\ell_{\Phi_{\exp}},\mathcal{H}}(h,x').
$$

*Additionally, for any hypothesis $h \in \mathcal{H}$, we have*

$$\mathcal{E}_{\mathsf{L}_{\Phi_{\exp}}}(h) - \mathcal{E}^*_{\mathsf{L}_{\Phi_{\exp}}}(\mathcal{H}) + \mathcal{M}_{\mathsf{L}_{\Phi_{\exp}}}(\mathcal{H}) \le 2\mathcal{E}_{\ell_{\Phi_{\exp}}}(h)\left(\mathcal{E}_{\ell_{\Phi_{\exp}}}(h) - \mathcal{E}^*_{\ell_{\Phi_{\exp}}}(\mathcal{H}) + \mathcal{M}_{\ell_{\Phi_{\exp}}}(\mathcal{H})\right).$$

**Proof** To simplify the notation, we will drop the dependency on $x$. Specifically, we use $\eta$ to denote $\eta(x)$, $\eta'$ to denote $\eta(x')$, $h$ to denote $h(x)$, and $h'$ to denote $h(x')$. Thus, we can write:

$$\Delta\overline{\mathcal{C}}_{\mathsf{L}_{\Phi_{\exp}},\mathcal{H}}(h,x,x') = \eta(1-\eta')e^{-h+h'} + \eta'(1-\eta)e^{-h'+h} - 2\sqrt{\eta(1-\eta')\eta'(1-\eta)}$$

$$\mathcal{C}_{\ell_{\Phi_{\exp}}}(h,x) = \eta e^{-h} + (1-\eta)e^{h}$$

$$\Delta\mathcal{C}_{\ell_{\Phi_{\exp}},\mathcal{H}}(h,x) = \eta e^{-h} + (1-\eta)e^{h} - 2\sqrt{\eta(1-\eta)}.$$

For any $A, B \in \mathbb{R}$, we have $(A+B)^2 \le 2(A^2 + B^2)$. To prove this inequality, observe that the function $x \mapsto x^2$ is convex. Therefore, we have:

$$(A+B)^2 = 4\left(\frac{A+B}{2}\right)^2 \le 4\left(\frac{A^2+B^2}{2}\right) = 2(A^2 + B^2).$$

In light of this inequality, we can write

$$\Delta\overline{\mathcal{C}}_{\mathsf{L}_{\Phi_{\exp}},\mathcal{H}}(h,x,x') = \left(\sqrt{\eta(1-\eta')e^{-h+h'}} - \sqrt{\eta'(1-\eta)e^{-h'+h}}\right)^2$$

$$= \left(\sqrt{(1-\eta')e^{h'}}\left(\sqrt{\eta e^{-h}} - \sqrt{(1-\eta)e^{h}}\right) + \sqrt{(1-\eta)e^{h}}\left(\sqrt{(1-\eta')e^{h'}} - \sqrt{\eta'e^{-h'}}\right)\right)^2$$

$$\le 2\left((1-\eta')e^{h'}\right)\left(\sqrt{\eta e^{-h}} - \sqrt{(1-\eta)e^{h}}\right)^2$$

$$\quad + 2\left((1-\eta)e^{h}\right)\left(\sqrt{\eta'e^{-h'}} - \sqrt{(1-\eta')e^{h'}}\right)^2 \quad ((A+B)^2 \le 2(A^2+B^2))$$

$$\Delta\overline{\mathcal{C}}_{\mathsf{L}_{\Phi_{\exp}},\mathcal{H}}(h,x,x') = \left(\sqrt{\eta'(1-\eta)e^{-h'+h}} - \sqrt{\eta(1-\eta')e^{-h+h'}}\right)^2$$

$$= \left(\sqrt{\eta'e^{-h'}}\left(\sqrt{(1-\eta)e^{h}} - \sqrt{\eta e^{-h}}\right) + \sqrt{\eta e^{-h}}\left(\sqrt{\eta'e^{-h'}} - \sqrt{(1-\eta')e^{h'}}\right)\right)^2$$

$$\le 2\left(\eta'e^{-h'}\right)\left(\sqrt{\eta e^{-h}} - \sqrt{(1-\eta)e^{h}}\right)^2$$

$$\quad + 2\left(\eta e^{-h}\right)\left(\sqrt{\eta'e^{-h'}} - \sqrt{(1-\eta')e^{h'}}\right)^2. \quad ((A+B)^2 \le 2(A^2+B^2))$$

Thus, by taking the mean of the two inequalities, we obtain:

$$\Delta\overline{\mathcal{C}}_{\mathsf{L}_{\Phi_{\exp}},\mathcal{H}}(h,x,x') \le \left(\eta'e^{-h'} + (1-\eta')e^{h'}\right)\left(\sqrt{\eta e^{-h}} - \sqrt{(1-\eta)e^{h}}\right)^2$$

$$\quad + \left(\eta e^{-h} + (1-\eta)e^{h}\right)\left(\sqrt{\eta'e^{-h'}} - \sqrt{(1-\eta')e^{h'}}\right)^2.$$

Therefore, we have

$$\Delta\overline{\mathcal{C}}_{\mathsf{L}_{\Phi_{\exp}},\mathcal{H}}(h,x,x') \le \mathcal{C}_{\ell_{\Phi_{\exp}}}(h,x')\Delta\mathcal{C}_{\ell_{\Phi_{\exp}},\mathcal{H}}(h,x) + \mathcal{C}_{\ell_{\Phi_{\exp}}}(h,x)\Delta\mathcal{C}_{\ell_{\Phi_{\exp}},\mathcal{H}}(h,x').$$

By Theorem 10, we obtain

$$\mathcal{E}_{\mathsf{L}_{\Phi_{\exp}}}(h) - \mathcal{E}^*_{\mathsf{L}_{\Phi_{\exp}}}(\mathcal{H}) + \mathcal{M}_{\mathsf{L}_{\Phi_{\exp}}}(\mathcal{H}) \le 2\mathcal{E}_{\ell_{\Phi_{\exp}}}(h)\left(\mathcal{E}_{\ell_{\Phi_{\exp}}}(h) - \mathcal{E}^*_{\ell_{\Phi_{\exp}}}(\mathcal{H}) + \mathcal{M}_{\ell_{\Phi_{\exp}}}(\mathcal{H})\right).$$

∎

### E.3. Proof of Theorem 11

**Theorem 11** *Assume that $\Phi$ is convex and differentiable, and satisfies $\Phi'(t) < 0$ for all $t \in \mathbb{R}$, and $\frac{\Phi'(t)}{\Phi'(-t)} = e^{-\nu t}$ for some $\nu > 0$. Then, $\ell_\Phi$ is calibrated with respect to $\mathsf{L}_\Phi$.*

**Proof** By the definition, we have

$$
\begin{aligned}
\mathcal{C}_\Phi(h, x) &= \eta(x)\Phi(h(x)) + (1 - \eta(x))\Phi(-h(x)) \\
\mathcal{C}_\Phi(h, x') &= \eta(x')\Phi(h(x')) + (1 - \eta(x'))\Phi(-h(x')) \\
\overline{\mathcal{C}}_{\mathsf{L}_\Phi}(h, x, x') &= \eta(x)(1 - \eta(x'))\Phi(h(x) - h(x')) + \eta(x')(1 - \eta(x))\Phi(-h(x) + h(x')).
\end{aligned}
$$

Therefore, by taking the derivative, we have

$$
\Delta\mathcal{C}_{\Phi,\mathcal{H}}(h, x) = 0 \implies \eta(x)\Phi'(h(x)) = (1 - \eta(x))\Phi'(-h(x)) \implies h(x) = \frac{1}{\nu}\log\left(\frac{\eta(x)}{1 - \eta(x)}\right)
$$

$$
\Delta\mathcal{C}_{\Phi,\mathcal{H}}(h, x') = 0 \implies \eta(x')\Phi'(h(x')) = (1 - \eta(x'))\Phi'(-h(x')) \implies h(x') = \frac{1}{\nu}\log\left(\frac{\eta(x')}{1 - \eta(x')}\right).
$$

Therefore, $h(x) - h(x') = \frac{1}{\nu}\log\left(\frac{\eta(x)(1-\eta(x'))}{\eta(x')(1-\eta(x))}\right)$. This satisfies that

$$
\eta(x)(1 - \eta(x'))\Phi'(h(x) - h(x')) = \eta(x')(1 - \eta(x))\Phi'(-h(x) + h(x')),
$$

which implies that $\Delta\overline{\mathcal{C}}_{\mathsf{L}_\Phi,\mathcal{H}}(h, x, x') = 0$ by taking the derivative. $\blacksquare$

### E.4. Proof of Theorem 13

**Theorem 13** *Assume that $\mathcal{H}$ is complete. For any $x$, define $u(x) = \max\{\eta(x), 1 - \eta(x)\}$. Then, the following inequality holds for the logistic loss $\Phi_{\log}$:*

$$
\Delta\overline{\mathcal{C}}_{\mathsf{L}_{\Phi_{\log}},\mathcal{H}}(h, x, x') \leq u(x')\Delta\mathcal{C}_{\ell_{\Phi_{\log}},\mathcal{H}}(h, x) + u(x)\Delta\mathcal{C}_{\ell_{\Phi_{\log}},\mathcal{H}}(h, x').
$$

*Furthermore, for any hypothesis $h \in \mathcal{H}$, we have*

$$
\mathcal{E}_{\mathsf{L}_{\Phi_{\log}}}(h) - \mathcal{E}^*_{\mathsf{L}_{\Phi_{\log}}}(\mathcal{H}) + \mathcal{M}_{\mathsf{L}_{\Phi_{\log}}}(\mathcal{H}) \leq 2\,\mathbb{E}[u(X)]\left(\mathcal{E}_{\ell_{\Phi_{\log}}}(h) - \mathcal{E}^*_{\ell_{\Phi_{\log}}}(\mathcal{H}) + \mathcal{M}_{\ell_{\Phi_{\log}}}(\mathcal{H})\right).
$$

**Proof** To simplify the notation, we will drop the dependency on $x$. Specifically, we use $\eta$ to denote $\eta(x)$, $\eta'$ to denote $\eta(x')$, $h$ to denote $h(x)$, and $h'$ to denote $h(x')$. Thus, we can write:

$$
\begin{aligned}
\Delta\overline{\mathcal{C}}_{\mathsf{L}_{\Phi_{\log}},\mathcal{H}}(h, x, x') &\leq \eta(1 - \eta')\log\left[1 + e^{-h+h'}\right] + \eta'(1 - \eta)\log\left[1 + e^{-h'+h}\right] \\
&\quad + \eta(1 - \eta')\log\left[\eta(1 - \eta')\right] + \eta'(1 - \eta)\log\left[\eta'(1 - \eta)\right] \\
\Delta\mathcal{C}_{\ell_{\Phi_{\log}},\mathcal{H}}(h, x) &= \eta\log\left[1 + e^{-h}\right] + (1 - \eta)\log\left[1 + e^{h}\right] \\
&\quad + \eta\log[\eta] + (1 - \eta)\log[(1 - \eta)].
\end{aligned}
$$

Let $\Phi_{\log}$ denote the logistic loss. $\Phi_{\log}$ is a convex function and for any $x$, we have $\Phi_{\log}(2x) \leq 2\Phi_{\log}(x)$. To prove this last inequality, observe that for any $x \in \mathbb{R}$, we have

$$\Phi_{\log}(2x) = \log(1 + e^{-2x}) \leq \log(1 + 2e^{-x} + e^{-2x}) = \log((1 + e^{-x})^2) = 2\log((1 + e^{-x})) = 2\Phi_{\log}(x).$$

Thus, we can write $\Phi_{\log}(h - h') = \Phi_{\log}(\frac{2h}{2} - \frac{2h'}{2}) \leq \frac{1}{2}(\Phi_{\log}(2h) + \Phi_{\log}(-2h')) \leq \Phi_{\log}(h) + \Phi_{\log}(-h')$. In light of this inequality, we can write

$$\Delta\overline{\mathcal{C}}_{\mathsf{L}_{\Phi_{\log}},\mathcal{H}}(h, x, x') \leq \eta(1 - \eta')(\Phi_{\log}(h) + \Phi_{\log}(-h')) + \eta'(1 - \eta)(\Phi_{\log}(h') + \Phi_{\log}(-h))$$
$$+ \eta(1 - \eta')\log[\eta(1 - \eta')] + \eta'(1 - \eta)\log[\eta'(1 - \eta)]$$
$$= \eta(1 - \eta')(\Phi_{\log}(h) + \Phi_{\log}(-h')) + \eta'(1 - \eta)(\Phi_{\log}(h') + \Phi_{\log}(-h))$$
$$+ \eta(1 - \eta')[\log\eta + \log(1 - \eta')] + \eta'(1 - \eta)[\log\eta' + \log(1 - \eta)].$$

Therefore, we have

$$\Delta\overline{\mathcal{C}}_{\mathsf{L}_{\Phi_{\log}},\mathcal{H}}(h, x, x') \leq \max\{\eta', 1 - \eta'\}\Delta\mathcal{C}_{\ell_{\Phi_{\log}},\mathcal{H}}(h, x) + \max\{\eta, 1 - \eta\}\Delta\mathcal{C}_{\ell_{\Phi_{\log}},\mathcal{H}}(h, x').$$

By Theorem 10, we obtain

$$\mathcal{E}_{\mathsf{L}_{\Phi_{\log}}}(h) - \mathcal{E}^*_{\mathsf{L}_{\Phi_{\log}}}(\mathcal{H}) + \mathcal{M}_{\mathsf{L}_{\Phi_{\log}}}(\mathcal{H})$$
$$\leq 2\,\mathbb{E}[\max\{\eta(x), (1 - \eta(x))\}]\left(\mathcal{E}_{\ell_{\Phi_{\log}}}(h) - \mathcal{E}^*_{\ell_{\Phi_{\log}}}(\mathcal{H}) + \mathcal{M}_{\ell_{\Phi_{\log}}}(\mathcal{H})\right).$$

$\blacksquare$

## E.5. Proof of Theorem 15

**Theorem 15 (Negative result for hinge losses)** *Assume that $\mathcal{H}$ contains the constant function* $1$. *For the hinge loss, if there exists a function pair* $(\Gamma_1, \Gamma_2)$ *such that the following holds for all $h \in \mathcal{H}$ and $(x, x') \in \mathcal{X} \times \mathcal{X}$, with some positive functions $\alpha_1: \mathcal{H} \times \mathcal{X} \to \mathbb{R}^*_+$ and $\alpha_2: \mathcal{H} \times \mathcal{X} \to \mathbb{R}^*_+$:*

$$\Delta\overline{\mathcal{C}}_{\mathsf{L}_{\Phi_{\text{hinge}}},\mathcal{H}}(h, x, x') \leq \Gamma_1\left(\alpha_1(h, x')\,\Delta\mathcal{C}_{\ell_{\Phi_{\text{hinge}}},\mathcal{H}}(h, x)\right) + \Gamma_2\left(\alpha_2(h, x)\,\Delta\mathcal{C}_{\ell_{\Phi_{\text{hinge}}},\mathcal{H}}(h, x')\right),$$

*then, we have* $\Gamma_1(0) + \Gamma_2(0) \geq \frac{1}{2}$.

**Proof** Consider the distribution that supports on $\{(x_0, x'_0)\}$. Let $1 \geq \eta(x_0) > \eta(x'_0) > \frac{1}{2}$, and $h_0 = 1 \in \mathcal{H}$. Then, for any $h \in \mathcal{H}$,

$$\mathcal{C}_{\ell_{\Phi_{\text{hinge}}}}(h, x_0) = \eta(x_0)\max\{0, 1 - h(x_0)\} + (1 - \eta(x_0))\max\{0, 1 + h(x_0)\} \geq 2(1 - \eta(x_0))$$
$$\mathcal{C}_{\ell_{\Phi_{\text{hinge}}}}(h, x'_0) = \eta(x'_0)\max\{0, 1 - h(x'_0)\} + (1 - \eta(x'_0))\max\{0, 1 + h(x'_0)\} \geq 2(1 - \eta(x'_0)),$$

where both equality can be achieved by $h_0 = 1$. Furthermore,

$$\overline{\mathcal{C}}_{\mathsf{L}_{\Phi_{\text{hinge}}}}(h_0, x_0, x'_0) = \eta(x_0)(1 - \eta(x'_0))\max\{0, 1 - h_0(x_0) + h_0(x'_0)\}$$
$$+ \eta(x'_0)(1 - \eta(x_0))\max\{1 - h_0(x'_0) + h_0(x_0)\}$$
$$= \eta(x_0)(1 - \eta(x'_0)) + \eta(x'_0)(1 - \eta(x_0))$$
$$\Delta\overline{\mathcal{C}}_{\mathsf{L}_{\Phi_{\text{hinge}}},\mathcal{H}}(h_0, x_0, x'_0) = \eta(x_0)(1 - \eta(x'_0)) + \eta(x'_0)(1 - \eta(x_0))$$
$$- 2\min\{\eta(x_0)(1 - \eta(x'_0)), \eta(x'_0)(1 - \eta(x_0))\}$$
$$= \eta(x_0) - \eta(x'_0).$$

Therefore, $\Delta \mathcal{C}_{\ell_{\Phi_{\text{hinge}}},\mathcal{H}}(h_0, x_0) = \Delta \mathcal{C}_{\ell_{\Phi_{\text{hinge}}},\mathcal{H}}(h_0, x_0') = 0$, but $\Delta \overline{\mathcal{C}}_{\mathsf{L}_{\Phi_{\text{hinge}}},\mathcal{H}}(h_0, x_0, x_0') \neq 0$, which implies that $\ell_{\Phi_{\text{hinge}}}$ is not $\mathcal{H}$-calibrated with respect to $\mathsf{L}_{\Phi_{\text{hinge}}}$.

Suppose that for all $h \in \mathcal{H}$, the following holds:

$$\Delta \overline{\mathcal{C}}_{\mathsf{L}_{\Phi_{\text{hinge}}},\mathcal{H}}(h, x_0, x_0') \leq \Gamma_1\Big(\alpha_1(h, x_0') \Delta \mathcal{C}_{\ell_{\Phi_{\text{hinge}}},\mathcal{H}}(h, x_0)\Big) + \Gamma_2\Big(\alpha_2(h, x_0) \Delta \mathcal{C}_{\ell_{\Phi_{\text{hinge}}},\mathcal{H}}(h, x_0')\Big).$$

Let $h = h_0$, then, for any $1 \geq \eta(x_0) > \eta(x_0') > \frac{1}{2}$, the following inequality holds:

$$\eta(x_0) - \eta(x_0') \leq \Gamma_1(0) + \Gamma_2(0).$$

This implies that $\Gamma_1(0) + \Gamma_2(0) \geq \frac{1}{2}$. ∎

## Appendix F. Generalization bounds

Here, we show that all our derived enhanced $\mathcal{H}$-consistency bounds can be used to provide novel enhanced generalization bounds in their respective settings.

### F.1. Standard multi-class classification

Let $S = ((x_1, y_1), \ldots, (x_m, y_m))$ be a finite sample drawn from $\mathcal{D}^m$. We denote by $\widehat{h}_S$ the minimizer of the empirical loss within $\mathcal{H}$ with respect to the constrained loss $\Phi^{\text{cstnd}}$:

$$\widehat{h}_S = \operatorname*{argmin}_{h \in \mathcal{H}} \widehat{\mathcal{E}}_{\Phi^{\text{cstnd}},S}(h) = \operatorname*{argmin}_{h \in \mathcal{H}} \frac{1}{m} \sum_{i=1}^{m} \Phi^{\text{cstnd}}(h, x_i, y_i).$$

Next, by using enhanced $\mathcal{H}$-consistency bounds for constrained losses $\Phi^{\text{cstnd}}$ in Theorem 5, we derive novel generalization bounds for the multi-class zero-one loss by upper bounding the surrogate estimation error $\mathcal{E}_{\Phi^{\text{cstnd}}}(\widehat{h}_S) - \mathcal{E}^*_{\Phi^{\text{cstnd}}}(\mathcal{H})$ with the complexity (e.g. the Rademacher complexity) of the family of functions associated with $\Phi^{\text{cstnd}}$ and $\mathcal{H}$: $\mathcal{H}_{\Phi^{\text{cstnd}}} = \{(x, y) \mapsto \Phi^{\text{cstnd}}(h, x, y) : h \in \mathcal{H}\}$.

Let $\mathfrak{R}_m^{\Phi^{\text{cstnd}}}(\mathcal{H})$ be the Rademacher complexity of $\mathcal{H}_{\Phi^{\text{cstnd}}}$ and $B_{\Phi^{\text{cstnd}}}$ an upper bound of the constrained loss $\Phi^{\text{cstnd}}$. The following generalization bound for the multi-class zero-one loss holds.

**Theorem 17 (Enhanced generalization bound with constrained losses)** *Assume that $\mathcal{H}$ is symmetric and complete. Then, the following generalization bound holds for $\widehat{h}_S$: for any $\delta > 0$, with probability at least $1 - \delta$ over the draw of an i.i.d sample $S$ of size $m$:*

$$\mathcal{E}_{\ell_{0-1}}(\widehat{h}_S) - \mathcal{E}^*_{\ell_{0-1}}(\mathcal{H}) + \mathcal{M}_{\ell_{0-1}}(\mathcal{H}) \leq \Gamma\left(4\mathfrak{R}_m^{\Phi^{\text{cstnd}}}(\mathcal{H}) + 2B_{\Phi^{\text{cstnd}}}\sqrt{\frac{\log \frac{2}{\delta}}{2m}} + \mathcal{M}_{\Phi^{\text{cstnd}}}(\mathcal{H})\right),$$

*where $\Gamma(x) = \dfrac{\sqrt{2}x^{\frac{1}{2}}}{\left(e^{\Lambda(\widehat{h}_S)}\right)^{\frac{1}{2}}}$ for $\Phi(u) = e^{-u}$, $\Gamma(x) = \dfrac{x}{1+\Lambda(\widehat{h}_S)}$ for $\Phi(u) = \max\{0, 1 - u\}$, and $\Gamma(x) = \dfrac{x^{\frac{1}{2}}}{1+\Lambda(\widehat{h}_S)}$ for $\Phi(u) = (1-u)^2 1_{u \leq 1}$. Additionally, $\Lambda(\widehat{h}_S) = \inf_{x \in \mathcal{X}} \max_{y \in \mathcal{Y}} \widehat{h}_S(x, y)$.*

**Proof** By using the standard Rademacher complexity bounds (Mohri et al., 2018), for any $\delta > 0$, with probability at least $1 - \delta$, the following holds for all $h \in \mathcal{H}$:

$$\left| \mathcal{E}_{\Phi^{\text{cstnd}}}(h) - \widehat{\mathcal{E}}_{\Phi^{\text{cstnd}},S}(h) \right| \leq 2\mathfrak{R}_m^{\Phi^{\text{cstnd}}}(\mathcal{H}) + B_{\Phi^{\text{cstnd}}} \sqrt{\frac{\log(2/\delta)}{2m}}.$$

Fix $\epsilon > 0$. By the definition of the infimum, there exists $h^* \in \mathcal{H}$ such that $\mathcal{E}_{\Phi^{\text{cstnd}}}(h^*) \leq \mathcal{E}_{\Phi^{\text{cstnd}}}^*(\mathcal{H}) + \epsilon$. By definition of $\widehat{h}_S$, we have

$$\begin{aligned}
&\mathcal{E}_{\Phi^{\text{cstnd}}}(\widehat{h}_S) - \mathcal{E}_{\Phi^{\text{cstnd}}}^*(\mathcal{H}) \\
&= \mathcal{E}_{\Phi^{\text{cstnd}}}(\widehat{h}_S) - \widehat{\mathcal{E}}_{\Phi^{\text{cstnd}},S}(\widehat{h}_S) + \widehat{\mathcal{E}}_{\Phi^{\text{cstnd}},S}(\widehat{h}_S) - \mathcal{E}_{\Phi^{\text{cstnd}}}^*(\mathcal{H}) \\
&\leq \mathcal{E}_{\Phi^{\text{cstnd}}}(\widehat{h}_S) - \widehat{\mathcal{E}}_{\Phi^{\text{cstnd}},S}(\widehat{h}_S) + \widehat{\mathcal{E}}_{\Phi^{\text{cstnd}},S}(h^*) - \mathcal{E}_{\Phi^{\text{cstnd}}}^*(\mathcal{H}) \\
&\leq \mathcal{E}_{\Phi^{\text{cstnd}}}(\widehat{h}_S) - \widehat{\mathcal{E}}_{\Phi^{\text{cstnd}},S}(\widehat{h}_S) + \widehat{\mathcal{E}}_{\Phi^{\text{cstnd}},S}(h^*) - \mathcal{E}_{\Phi^{\text{cstnd}}}^*(h^*) + \epsilon \\
&\leq 2\left[ 2\mathfrak{R}_m^{\Phi^{\text{cstnd}}}(\mathcal{H}) + B_{\Phi^{\text{cstnd}}} \sqrt{\frac{\log(2/\delta)}{2m}} \right] + \epsilon.
\end{aligned}$$

Since the inequality holds for all $\epsilon > 0$, it implies:

$$\mathcal{E}_{\Phi^{\text{cstnd}}}(\widehat{h}_S) - \mathcal{E}_{\Phi^{\text{cstnd}}}^*(\mathcal{H}) \leq 4\mathfrak{R}_m^{\Phi^{\text{cstnd}}}(\mathcal{H}) + 2B_{\Phi^{\text{cstnd}}} \sqrt{\frac{\log(2/\delta)}{2m}}.$$

Plugging in this inequality in the bounds of Theorem 5 completes the proof. ∎

To the best of our knowledge, Theorem 17 provides the first enhanced finite-sample guarantees, expressed in terms of minimizability gaps, for the estimation error of the minimizer of constrained losses with respect to the multi-class zero-one loss, incorporating a quantity $\Lambda(\widehat{h}_S)$ depending on $\widehat{h}_S$. The proof uses our enhanced $\mathcal{H}$-consistency bounds for constrained losses (Theorem 5), as well as standard Rademacher complexity guarantees.

## F.2. Classification under low-noise conditions

Let $S = ((x_1, y_1), \ldots, (x_m, y_m))$ be a finite sample drawn from $\mathcal{D}^m$. We denote by $\widehat{h}_S$ the minimizer of the empirical loss within $\mathcal{H}$ with respect to a surrogate loss $\ell$: $\widehat{h}_S = \operatorname{argmin}_{h \in \mathcal{H}} \widehat{\mathcal{E}}_{\ell,S}(h) = \operatorname{argmin}_{h \in \mathcal{H}} \frac{1}{m} \sum_{i=1}^{m} \ell(h, x_i, y_i)$.

Next, by using enhanced $\mathcal{H}$-consistency bounds for surrogate losses $\ell$ in Theorems 6 and 9, we derive novel generalization bounds for the binary and multi-class zero-one loss under low-noise conditions, by upper bounding the surrogate estimation error $\mathcal{E}_\ell(\widehat{h}_S) - \mathcal{E}_\ell^*(\mathcal{H})$ with the complexity (e.g. the Rademacher complexity) of the family of functions associated with $\ell$ and $\mathcal{H}$: $\mathcal{H}_\ell = \{(x, y) \mapsto \ell(h, x, y) : h \in \mathcal{H}\}$.

Let $\mathfrak{R}_m^\ell(\mathcal{H})$ be the Rademacher complexity of $\mathcal{H}_\ell$ and $B_\ell$ an upper bound of the surrogate loss $\ell$. The following generalization bounds for the binary and multi-class zero-one loss hold.

**Theorem 18 (Enhanced binary generalization bound under the Tsybakov noise assumption)** *Consider a binary classification setting where the Tsybakov noise assumption holds. Assume that the following holds for all $h \in \mathcal{H}$ and $x \in \mathcal{X}$: $\Delta\mathcal{C}_{\ell_{0-1}^{\mathrm{bi}}, \mathcal{H}}(h, x) \leq \Gamma(\Delta\mathcal{C}_{\ell, \mathcal{H}}(h, x))$, with $\Gamma(x) = x^{\frac{1}{s}}$, for some $s \geq 1$. Then, for any $h \in \mathcal{H}$,*

$$\mathcal{E}_{\ell_{0-1}^{\mathrm{bi}}}(\widehat{h}_S) - \mathcal{E}_{\ell_{0-1}^{\mathrm{bi}}}^*(\mathcal{H}) \leq c^{\frac{s-1}{s-\alpha(s-1)}} \left[ 4\mathfrak{R}_m^\ell(\mathcal{H}) + 2B_\ell \sqrt{\frac{\log(2/\delta)}{2m}} + \mathcal{M}_\ell(\mathcal{H}) \right]^{\frac{1}{s-\alpha(s-1)}}.$$

**Proof** By using the standard Rademacher complexity bounds (Mohri et al., 2018), for any $\delta > 0$, with probability at least $1 - \delta$, the following holds for all $h \in \mathcal{H}$:

$$\left| \mathcal{E}_\ell(h) - \widehat{\mathcal{E}}_{\ell,S}(h) \right| \leq 2\mathfrak{R}_m^\ell(\mathcal{H}) + B_\ell \sqrt{\frac{\log(2/\delta)}{2m}}.$$

Fix $\epsilon > 0$. By the definition of the infimum, there exists $h^* \in \mathcal{H}$ such that $\mathcal{E}_\ell(h^*) \leq \mathcal{E}_\ell^*(\mathcal{H}) + \epsilon$. By definition of $\widehat{h}_S$, we have

$$\begin{aligned}
\mathcal{E}_\ell(\widehat{h}_S) &- \mathcal{E}_\ell^*(\mathcal{H}) \\
&= \mathcal{E}_\ell(\widehat{h}_S) - \widehat{\mathcal{E}}_{\ell,S}(\widehat{h}_S) + \widehat{\mathcal{E}}_{\ell,S}(\widehat{h}_S) - \mathcal{E}_\ell^*(\mathcal{H}) \\
&\leq \mathcal{E}_\ell(\widehat{h}_S) - \widehat{\mathcal{E}}_{\ell,S}(\widehat{h}_S) + \widehat{\mathcal{E}}_{\ell,S}(h^*) - \mathcal{E}_\ell^*(\mathcal{H}) \\
&\leq \mathcal{E}_\ell(\widehat{h}_S) - \widehat{\mathcal{E}}_{\ell,S}(\widehat{h}_S) + \widehat{\mathcal{E}}_{\ell,S}(h^*) - \mathcal{E}_\ell^*(h^*) + \epsilon \\
&\leq 2 \left[ 2\mathfrak{R}_m^\ell(\mathcal{H}) + B_\ell \sqrt{\frac{\log(2/\delta)}{2m}} \right] + \epsilon.
\end{aligned}$$

Since the inequality holds for all $\epsilon > 0$, it implies:

$$\mathcal{E}_\ell(\widehat{h}_S) - \mathcal{E}_\ell^*(\mathcal{H}) \leq 4\mathfrak{R}_m^\ell(\mathcal{H}) + 2B_\ell \sqrt{\frac{\log(2/\delta)}{2m}}.$$

Plugging in this inequality in the bounds of Theorem 6 completes the proof. ∎

**Theorem 19 (Enhanced multi-class generalization bound under the Tsybakov noise assumption)** *Consider a multi-class classification setting where the Tsybakov noise assumption holds. Assume that the following holds for all $h \in \mathcal{H}$ and $x \in \mathcal{X}$: $\Delta\mathcal{C}_{\ell_{0-1}, \mathcal{H}}(h, x) \leq \Gamma(\Delta\mathcal{C}_{\ell, \mathcal{H}}(h, x))$, with $\Gamma(x) = x^{\frac{1}{s}}$, for some $s \geq 1$. Then, for any $h \in \mathcal{H}$,*

$$\mathcal{E}_{\ell_{0-1}}(\widehat{h}_S) - \mathcal{E}_{\ell_{0-1}}^*(\mathcal{H}) \leq c^{\frac{s-1}{s-\alpha(s-1)}} \left[ 4\mathfrak{R}_m^\ell(\mathcal{H}) + 2B_\ell \sqrt{\frac{\log(2/\delta)}{2m}} + \mathcal{M}_\ell(\mathcal{H}) \right]^{\frac{1}{s-\alpha(s-1)}}.$$

**Proof** By using the standard Rademacher complexity bounds (Mohri et al., 2018), for any $\delta > 0$, with probability at least $1 - \delta$, the following holds for all $h \in \mathcal{H}$:

$$\left|\mathcal{E}_\ell(h) - \widehat{\mathcal{E}}_{\ell,S}(h)\right| \leq 2\mathfrak{R}_m^\ell(\mathcal{H}) + B_\ell\sqrt{\frac{\log(2/\delta)}{2m}}.$$

Fix $\epsilon > 0$. By the definition of the infimum, there exists $h^* \in \mathcal{H}$ such that $\mathcal{E}_\ell(h^*) \leq \mathcal{E}_\ell^*(\mathcal{H}) + \epsilon$. By definition of $\widehat{h}_S$, we have

$$\begin{aligned}
\mathcal{E}_\ell(\widehat{h}_S) &- \mathcal{E}_\ell^*(\mathcal{H}) \\
&= \mathcal{E}_\ell(\widehat{h}_S) - \widehat{\mathcal{E}}_{\ell,S}(\widehat{h}_S) + \widehat{\mathcal{E}}_{\ell,S}(\widehat{h}_S) - \mathcal{E}_\ell^*(\mathcal{H}) \\
&\leq \mathcal{E}_\ell(\widehat{h}_S) - \widehat{\mathcal{E}}_{\ell,S}(\widehat{h}_S) + \widehat{\mathcal{E}}_{\ell,S}(h^*) - \mathcal{E}_\ell^*(\mathcal{H}) \\
&\leq \mathcal{E}_\ell(\widehat{h}_S) - \widehat{\mathcal{E}}_{\ell,S}(\widehat{h}_S) + \widehat{\mathcal{E}}_{\ell,S}(h^*) - \mathcal{E}_\ell^*(h^*) + \epsilon \\
&\leq 2\left[2\mathfrak{R}_m^\ell(\mathcal{H}) + B_\ell\sqrt{\frac{\log(2/\delta)}{2m}}\right] + \epsilon.
\end{aligned}$$

Since the inequality holds for all $\epsilon > 0$, it implies:

$$\mathcal{E}_\ell(\widehat{h}_S) - \mathcal{E}_\ell^*(\mathcal{H}) \leq 4\mathfrak{R}_m^\ell(\mathcal{H}) + 2B_\ell\sqrt{\frac{\log(2/\delta)}{2m}}.$$

Plugging in this inequality in the bounds of Theorem 9 completes the proof. ∎

To the best of our knowledge, Theorems 18 and 19 provide the first enhanced finite-sample guarantees, expressed in terms of minimizability gaps, for the estimation error of the minimizer of surrogate losses with respect to the binary and multi-class zero-one loss, under the Tsybakov noise assumption. The proofs use our enhanced $\mathcal{H}$-consistency bounds (Theorems 6 and 9), as well as standard Rademacher complexity guarantees.

Note that our enhanced $\mathcal{H}$-consistency bounds can also be combined with standard bounds of $\mathcal{E}_\ell(\widehat{h}_S) - \mathcal{E}_\ell^*(\mathcal{H})$ in the case of Tsybakov noise, which can yield a fast rate. For example, our enhanced $\mathcal{H}$-consistency bounds in binary classification (Theorems 6) can be combined with the fast estimation rates described in (Bartlett et al., 2006, Section 4), which can lead to enhanced finite-sample guarantees in the case of Tsybakov noise. This is even true in the case of $\mathcal{H} = \mathcal{H}_{\text{all}}$, with a more favorable factor of one instead of $2^{\frac{-s}{s-\alpha(s-1)}}$.

### F.3. Bipartite ranking

Let $S = ((x_1, y_1), (x_1', y_1'), \ldots, (x_m, y_m), (x_m', y_m'))$ be a finite sample drawn from $(\mathcal{D} \times \mathcal{D})^m$. We denote by $\widehat{h}_S$ the minimizer of the empirical loss within $\mathcal{H}$ with respect to a classification surrogate loss $\ell_\Phi$: $\widehat{h}_S = \operatorname{argmin}_{h \in \mathcal{H}} \widehat{\mathcal{E}}_{\ell_\Phi,S}(h) = \operatorname{argmin}_{h \in \mathcal{H}} \frac{1}{m}\sum_{i=1}^m \ell_\Phi(h, x_i, y_i)$.

Next, by using enhanced $\mathcal{H}$-consistency bounds for classification surrogate losses $\ell_\Phi$ in Theorems 12 and 13, we derive novel generalization bounds for bipartite ranking surrogate losses $\mathsf{L}_\Phi$, by upper bounding the surrogate estimation error $\mathcal{E}_{\ell_\Phi}(\widehat{h}_S) - \mathcal{E}_{\ell_\Phi}^*(\mathcal{H})$ with the complexity (e.g. the Rademacher complexity) of the family of functions associated with $\ell_\Phi$ and $\mathcal{H}$: $\mathcal{H}_{\ell_\Phi} = \{(x, y) \mapsto \ell_\Phi(h, x, y) : h \in \mathcal{H}\}$.

Let $\mathfrak{R}_m^{\ell_\Phi}(\mathcal{H})$ be the Rademacher complexity of $\mathcal{H}_{\ell_\Phi}$ and $B_{\ell_\Phi}$ an upper bound of the classification surrogate $\ell_\Phi$. The following generalization bounds for the bipartite ranking surrogate losses $\mathsf{L}_\Phi$ hold.

**Theorem 20 (Enhanced generalization bound with AdaBoost)** *Assume that $\mathcal{H}$ is complete. Then, for any hypothesis $h \in \mathcal{H}$, we have*

$$\mathcal{E}_{\mathsf{L}_{\Phi_{\exp}}}(\widehat{h}_S) - \mathcal{E}^*_{\mathsf{L}_{\Phi_{\exp}}}(\mathcal{H}) + \mathcal{M}_{\mathsf{L}_{\Phi_{\exp}}}(\mathcal{H})$$

$$\leq 2\mathcal{E}_{\ell_{\Phi_{\exp}}}(\widehat{h}_S)\left(4\mathfrak{R}_m^{\ell_{\Phi_{\exp}}}(\mathcal{H}) + 2B_{\ell_{\Phi_{\exp}}}\sqrt{\frac{\log(2/\delta)}{2m}} + \mathcal{M}_{\ell_{\Phi_{\exp}}}(\mathcal{H})\right).$$

**Proof** By using the standard Rademacher complexity bounds (Mohri et al., 2018), for any $\delta > 0$, with probability at least $1 - \delta$, the following holds for all $h \in \mathcal{H}$:

$$\left|\mathcal{E}_{\ell_{\Phi_{\exp}}}(h) - \widehat{\mathcal{E}}_{\ell_{\Phi_{\exp}}, S}(h)\right| \leq 2\mathfrak{R}_m^{\ell_{\Phi_{\exp}}}(\mathcal{H}) + B_{\ell_{\Phi_{\exp}}}\sqrt{\frac{\log(2/\delta)}{2m}}.$$

Fix $\epsilon > 0$. By the definition of the infimum, there exists $h^* \in \mathcal{H}$ such that $\mathcal{E}_{\ell_{\Phi_{\exp}}}(h^*) \leq \mathcal{E}^*_{\ell_{\Phi_{\exp}}}(\mathcal{H}) + \epsilon$. By definition of $\widehat{h}_S$, we have

$$\mathcal{E}_{\ell_{\Phi_{\exp}}}(\widehat{h}_S) - \mathcal{E}^*_{\ell_{\Phi_{\exp}}}(\mathcal{H})$$

$$= \mathcal{E}_{\ell_{\Phi_{\exp}}}(\widehat{h}_S) - \widehat{\mathcal{E}}_{\ell_{\Phi_{\exp}}, S}(\widehat{h}_S) + \widehat{\mathcal{E}}_{\ell_{\Phi_{\exp}}, S}(\widehat{h}_S) - \mathcal{E}^*_{\ell_{\Phi_{\exp}}}(\mathcal{H})$$

$$\leq \mathcal{E}_{\ell_{\Phi_{\exp}}}(\widehat{h}_S) - \widehat{\mathcal{E}}_{\ell_{\Phi_{\exp}}, S}(\widehat{h}_S) + \widehat{\mathcal{E}}_{\ell_{\Phi_{\exp}}, S}(h^*) - \mathcal{E}^*_{\ell_{\Phi_{\exp}}}(\mathcal{H})$$

$$\leq \mathcal{E}_{\ell_{\Phi_{\exp}}}(\widehat{h}_S) - \widehat{\mathcal{E}}_{\ell_{\Phi_{\exp}}, S}(\widehat{h}_S) + \widehat{\mathcal{E}}_{\ell_{\Phi_{\exp}}, S}(h^*) - \mathcal{E}^*_{\ell_{\Phi_{\exp}}}(h^*) + \epsilon$$

$$\leq 2\left[2\mathfrak{R}_m^{\ell_{\Phi_{\exp}}}(\mathcal{H}) + B_{\ell_{\Phi_{\exp}}}\sqrt{\frac{\log(2/\delta)}{2m}}\right] + \epsilon.$$

Since the inequality holds for all $\epsilon > 0$, it implies:

$$\mathcal{E}_{\ell_{\Phi_{\exp}}}(\widehat{h}_S) - \mathcal{E}^*_{\ell_{\Phi_{\exp}}}(\mathcal{H}) \leq 4\mathfrak{R}_m^{\ell_{\Phi_{\exp}}}(\mathcal{H}) + 2B_{\ell_{\Phi_{\exp}}}\sqrt{\frac{\log(2/\delta)}{2m}}.$$

Plugging in this inequality in the bounds of Theorem 12 completes the proof. ∎

**Theorem 21 (Enhanced generalization bound with logistic regression)** *Assume that $\mathcal{H}$ is complete. For any $x$, define $u(x) = \max\{\eta(x), 1 - \eta(x)\}$. Then, for any hypothesis $h \in \mathcal{H}$, we have*

$$\mathcal{E}_{\mathsf{L}_{\Phi_{\log}}}(\widehat{h}_S) - \mathcal{E}^*_{\mathsf{L}_{\Phi_{\log}}}(\mathcal{H}) + \mathcal{M}_{\mathsf{L}_{\Phi_{\log}}}(\mathcal{H})$$

$$\leq 2\,\mathbb{E}[u(X)]\left(4\mathfrak{R}_m^{\ell_{\Phi_{\log}}}(\mathcal{H}) + 2B_{\ell_{\Phi_{\log}}}\sqrt{\frac{\log(2/\delta)}{2m}} + \mathcal{M}_{\ell_{\Phi_{\log}}}(\mathcal{H})\right).$$

**Proof** By using the standard Rademacher complexity bounds (Mohri et al., 2018), for any $\delta > 0$, with probability at least $1 - \delta$, the following holds for all $h \in \mathcal{H}$:

$$\left|\mathcal{E}_{\ell_{\Phi_{\log}}}(h) - \widehat{\mathcal{E}}_{\ell_{\Phi_{\log}}, S}(h)\right| \leq 2\mathfrak{R}_m^{\ell_{\Phi_{\log}}}(\mathcal{H}) + B_{\ell_{\Phi_{\log}}}\sqrt{\frac{\log(2/\delta)}{2m}}.$$

Fix $\epsilon > 0$. By the definition of the infimum, there exists $h^* \in \mathcal{H}$ such that $\mathcal{E}_{\ell_{\Phi_{\log}}}(h^*) \leq \mathcal{E}^*_{\ell_{\Phi_{\log}}}(\mathcal{H}) + \epsilon$. By definition of $\widehat{h}_S$, we have

$$\mathcal{E}_{\ell_{\Phi_{\log}}}(\widehat{h}_S) - \mathcal{E}^*_{\ell_{\Phi_{\log}}}(\mathcal{H})$$

$$= \mathcal{E}_{\ell_{\Phi_{\log}}}(\widehat{h}_S) - \widehat{\mathcal{E}}_{\ell_{\Phi_{\log}},S}(\widehat{h}_S) + \widehat{\mathcal{E}}_{\ell_{\Phi_{\log}},S}(\widehat{h}_S) - \mathcal{E}^*_{\ell_{\Phi_{\log}}}(\mathcal{H})$$

$$\leq \mathcal{E}_{\ell_{\Phi_{\log}}}(\widehat{h}_S) - \widehat{\mathcal{E}}_{\ell_{\Phi_{\log}},S}(\widehat{h}_S) + \widehat{\mathcal{E}}_{\ell_{\Phi_{\log}},S}(h^*) - \mathcal{E}^*_{\ell_{\Phi_{\log}}}(\mathcal{H})$$

$$\leq \mathcal{E}_{\ell_{\Phi_{\log}}}(\widehat{h}_S) - \widehat{\mathcal{E}}_{\ell_{\Phi_{\log}},S}(\widehat{h}_S) + \widehat{\mathcal{E}}_{\ell_{\Phi_{\log}},S}(h^*) - \mathcal{E}^*_{\ell_{\Phi_{\log}}}(h^*) + \epsilon$$

$$\leq 2\left[ 2\mathfrak{R}_m^{\ell_{\Phi_{\log}}}(\mathcal{H}) + B_{\ell_{\Phi_{\log}}}\sqrt{\frac{\log(2/\delta)}{2m}} \right] + \epsilon.$$

Since the inequality holds for all $\epsilon > 0$, it implies:

$$\mathcal{E}_{\ell_{\Phi_{\log}}}(\widehat{h}_S) - \mathcal{E}^*_{\ell_{\Phi_{\log}}}(\mathcal{H}) \leq 4\mathfrak{R}_m^{\ell_{\Phi_{\log}}}(\mathcal{H}) + 2B_{\ell_{\Phi_{\log}}}\sqrt{\frac{\log(2/\delta)}{2m}}.$$

Plugging in this inequality in the bounds of Theorem 13 completes the proof. ∎

To the best of our knowledge, Theorems 20 and 21 provide the first enhanced finite-sample guarantees, expressed in terms of minimizability gaps, for the estimation error of the minimizer of classification surrogate losses with respect to the bipartite ranking surrogate losses. Theorem 20 is remarkable since it provide finite simple bounds of the estimation error of the RankBoost loss function by that of AdaBoost. Significantly, Theorems 21 implies a parallel finding for logistic regression analogous to that of AdaBoost. The proofs use our enhanced $\mathcal{H}$-consistency bounds (Theorems 12 and 13), as well as standard Rademacher complexity guarantees.

