# OpenReview forum: "Enhanced $H$-Consistency Bounds"
_algorithmiclearningtheory.org/ALT/2025/Conference — ALT 2025_

### Official Review · Reviewer_Tvdo · 2024-10-14
**Need more comparison with the previous consistency bounds**

**Rating:** 6
**Confidence:** 3

**Review:**

In this submission, the authors introduced novel tools for deriving enhanced $H$-consistency bounds in various learning settings, including multi-class classification, low-noise regimes, and bipartite ranking. They established substantially more favorable guarantees for several settings and demonstrated unexpected connections between classification and bipartite ranking performances for the exponential and logistic losses. In general, finer and more favorable $H$-consistency bounds are established.

The authors have clearly stated the new enhanced $H$-consistency bounds as well as its applications to different scenarios. The structure is clear and the authors provided all the detailed mathematical proof for all the theoretical analysis.

The only weakness and suggestions is that the authors should explain more about the difference or improvement between the proposed enhanced $H$-consistency bounds in this submission and the previous $H$-consistency bounds in the related works. Theorem 1 and theorem 2 generalize previous results from (Awasthi et al., 2022a,b), which can be recovered as special cases when $\alpha = 1$ and $\beta = 1$. The authors argue that compared to earlier approaches, these new tools offer more precise $H$-consistency bounds in familiar settings and extend them to new scenarios where previous methods are insufficient.

Could the authors explain more about the reason why earlier approaches didn't work well for some settings and previous methods can not be applied to some scenarios? The new enhanced $H$-consistency is applied to multi-class classification, binary classification, and bipartite ranking with exponential loss, logistic loss and hinge loss. The authors should make a detailed comparison between the applications of the earlier approaches to these setting and the proposed enhanced $H$-consistency to these scenarios. The authors should provide the counter parts of Theorem 5 to 12 with applications of the previous $H$-consistency. Through this comparison, we can see the improvements of the novel method in this submission and significantly validate the argument made by the authors.

**Paper Award:**

No

---

> ### Author Response · Authors · 2024-11-22
>
> We appreciate the reviewer's positive comments and their useful comments.
>
> To address their main question: yes, previous $H$-consistency bounds methods (Awasthi et al., 2022a,b; Mao et al., 2023e,b,d) can indeed be applied in the same settings or scenarios as ours. However, the $H$-consistency bounds provided by prior work are distribution-agnostic, meaning they hold for any distribution and their form remains invariant with respect to both the distribution and the hypothesis $h$. Specifically, these bounds rely on a fixed convex function $\Psi$ or concave function $\Gamma$, which is independent of both the distribution and the hypothesis.
>
> As a consequence, earlier $H$-consistency bounds cannot leverage beneficial assumptions such as the low-noise condition. For example, the bounds for smooth loss functions in prior work exhibit a square-root behavior (regardless of the distribution), whereas our bounds achieve a linear rate under the Massart noise condition, and an intermediate rate between linear and square-root for various values of $\alpha$ under the Tsybakov noise condition.
>
> Similarly, prior work does not take advantage of favorable $h$-dependent quantities, unlike our more refined bipartite ranking bounds. To make this distinction clearer, we will include a direct comparison of previous bounds in the same settings, as suggested by the reviewer.
>
> Additionally, in the case of bipartite ranking, the previous methods apply only to the logistic loss. For the exponential loss, however, there is a non-constant factor that can grow arbitrarily large if the predictor deviates significantly from the best-in-class hypothesis. Consequently, the previous methods cannot be applied in this case and do not yield a bound.
>
> The key technical distinction in our work lies in the introduction of the $\alpha$ and $\beta$ functions, which allow us to derive enhanced bounds with a non-constant factor $\gamma$, capturing properties of the distribution and the predictor $h$.
>
> Finally, we want to emphasize that the impact of our results goes beyond the improved $H$-consistency bounds. Our contribution includes novel fundamental tools with wide applicability. For instance, our methods can be adapted to comp-sum losses, leading to improved adversarial robustness algorithms, similar to the advances in adversarial algorithms by Mao et al. (2023e). Moreover, these tools can connect existing algorithms, shedding light on empirical observations, such as AdaBoost’s strong ranking performance, which aligns with similar findings for the logistic loss.

---

### Official Review · Reviewer_pL3N · 2024-11-08
**Well-written, clear contributions**

**Rating:** 7
**Confidence:** 2

**Review:**

The paper focuses on learning with surrogate losses. More precisely, they revisit the H-consistency setting, and prove more refined results where the bounds now can depend on the choice of the particular hypothesis. They instantiate their results on three different problems: binary and multi-class classification, and bipartite ranking.

To start with, I am not an expert on learning with surrogate losses, hence some of the concepts presented in the paper were new to me. However, the paper is very-well written and easy to follow, and it is quite clear what the improvements that the authors brought to the current literature. By looking at the proofs, it is not obvious to me if the authors developed new proof techniques (it seemed to me they used quite standard tools), yet I think the results they proved seem significant. In the classification cases they refine existing results, whereas (I think it is the more interesting part) in the bipartite ranking problem their results provide more theoretical support for explaining existing empirical observations.

In general, the paper has clear improvements over the prior work, it is well-written, and sheds more light to certain empirical observations. Hence, given these explanations and the fact that I am not an expert in this field, I recommend an accept with a low confidence score.

**Paper Award:**

No

---

> ### Author Response · Authors · 2024-11-22
>
> We appreciate the reviewer’s positive feedback on the significance of our results and the quality of our writing. They have a very good understanding of our main contributions, and we fully agree with their observations.
>
> Below, we simply comment on our proofs.  The proofs of our core tools are indeed novel and more complex than their counterparts in prior work due to the presence of the functions $\alpha$ and $\beta$. Our proof technique involves a tailored application of Jensen’s inequality leveraging function $\beta$, the use of Hölder’s inequality adapted for the $\alpha$ function, and the application of the FKG Inequality. Remarkably, these steps enable us to derive guarantees which admit lower bounds.
>
> The proofs of our enhanced bounds require careful case-by-case analysis to relate the conditional regrets between various surrogate losses and the target losses through appropriate functions $\alpha$ and $\beta$, which differ significantly from the proofs in previous work on $H$-consistency bounds.
>
> For example, in the proof of Theorem 13, we first established and leveraged the sub-additivity of $\Phi_{\log}$: $\Phi_{\mathrm{log}}(h - h') \leq
> \Phi_{\mathrm{log}}(h) + \Phi_{\mathrm{log}}(-h')$, to constructively derive an upper
> bound for the conditional regret of bipartite misranking logistic loss
>  $\Delta \overline{\mathcal{C}} _{\mathsf{L} _{\Phi _{\mathrm{log}}},  \mathcal{H}}(h, x, x')$,  in terms of the conditional regrets of classification logistic loss $\Delta \mathcal{C} _{\ell _{\Phi _{\mathrm{log}}}, \mathcal{H}}(h, x)$ and
> $\Delta \mathcal{C} _{\ell _{\Phi _{\mathrm{log}}}, \mathcal{H}}(h, x')$.
>
> We then applied our newly developed tool, Theorem 10, with $\alpha_1(h, x') = \max \\{\eta(x'), 1 - \eta(x') \\}$ and $\alpha_2(h, x) = \max \\{\eta(x), 1 - \eta(x) \\}$. Our new tools facilitated the derivation of these non-trivial inequalities where non-constant factors play a crucial role.
>
> In the final version of the paper, we will provide a more comprehensive explanation of the innovative aspects of our proof techniques.

---

### Official Review · Reviewer_dSac · 2024-11-10
**Weak accept; interesting problem, but paper reads like a list of results with no intuition or proof sketches**

**Rating:** 6
**Confidence:** 2

**Review:**

## Paper Summary
The goal of $\mathcal{H}$-consistency bounds are to relate the performance of a
classifier $h \in \mathcal{H}$ under a surrogate loss function to its
performance with respect to the target 0-1 loss. In particular, the goal is to
bound the 0-1 generalization error that is in excess of the best-in-class. This
is to be given as a function of the excess surrogate generalization error:

$$\mathrm{error}_{01}(h) - \min_{h' \in \mathcal{H}} \mathrm{error}_{01}(h')  \leq \mathrm{function}\left(\mathrm{error}_{\mathrm{surrogate}}(h) - \min_{h' \in \mathcal{H}}\, \mathrm{error}_{\mathrm{surrogate}}(h')\right).$$

This paper provides tools that allow for more fine-grained bounds that achieve
rates that are not achievable by prior work. There are settings for which these
bounds are tight. This paper also applies these tools to show
$\mathcal{H}$-consistency bounds for various settings.

## Review
This paper addresses a very interesting question and seems to develop some
fairly useful and novel tools for $\mathcal{H}$-consistency. The main weakness
of this paper is its presentation of ideas, which are a little hard to follow.
While the writing itself is nice, it does not help ease the reader into the
technical details. For example, the tools are presented in its technical form
almost immediately with insufficient exposition/intuition (e.g. where did
$\alpha$ and $\beta$ come from? why is there a need for instance-dependence?).
After the tools are presented, they are applied for a seemingly disconnected
collection of settings (multiclass, noisy, and pairwise rankings).

I think it would make the paper muh more accessible if the authors devoted more
time to motivating the new "fundamental tools". Perhaps a simple example to
demonstrate how these new techniques are able to achieve more optimal bounds. In
fact, it might even be worth just focusing on one of the settings in the main
body of the paper (probably the noisy setting), and helping the reader develop
the intuition for how the stronger bounds were derived. The proof of Theorem 1,
which is nice and short, could for example be even moved to the main body and
given more exposition.

Perhaps some of the following questions could be helpful for revising the
presentation.

1. I didn't understand the story about the faster rates. The authors write
   "recent work by Mao et. al. (2024a) shows that for all smoothed surrogate
   losses in binary classification, $\Gamma(\epsilon)$ behaves as
   $\sqrt{\epsilon}$ near zero". And then, later, "we will show that under
   certain noise conditions in classification, the behavior of $\Gamma$ can
   outperform the typical square-root dependence, approaching near-linear
   behavior". What does this mean? Is this saying that the earlier result was
   wrong because it is possible to achieve linear rates? Or, is it saying that
   the earlier analysis is not always tight? This comparison sounds interesting,
   but I did not understand what the specific differences are. Is it under more
   benign noise that faster rates are obtained? What about the new tools enable
   a tighter analysis?

2. Are there intuition behind $\alpha$, $\beta$, and $\gamma$? They really seem
   to appear out of nowhere to make things work out.

3. Minor thing: what is "approximation error" (e.g. page 4, first new
   paragraph)? It is not defined. Perhaps it is
   $\mathcal{E}_{\ell}^*(\mathcal{H})-\mathcal{E}_{\ell}^*(\mathcal{H}_{\mathrm{all}})$?

4. Are there other takeaways that should be emphasized from the results in the
   multiclass, noisy, ranking settings? They seem like they are a nice series of
   results, but they are a bit inscrutable to me; perhaps the lack of
   presentation is doing these results a disservice.

Overall, this paper lists many results, but essentially no proof sketches. And
so, the results seem believable, but it is fairly difficult for someone not
working closely in the area to appreciate them. The problem seems important and
the stated contributions, if correct, are good. I was not able to closely look
at the appendix and did not verify the proofs (and I didn't know how to do
high-level sanity checks for these results). I think it is well-worth to give
some thought to making the ideas more accessible. Overall, the writing style is
good and the general motivation is stated well.

I chose 6-weak accept because the technical results seem like good contributions if correct. However, my confidence is 2: while I would like to verify the proofs, there is too much overhead because there's not really an entry point for someone who doesn't already work on H-consistency.

**Paper Award:**

No

---

> ### Author Response · Authors · 2024-11-22
>
> Thank you for your suggestions. We will certainly include more discussions on intuitions and insights related to our enhanced bounds and the applicability of our fundamental tools, leveraging additional pages in the final version.
>
> Here are responses to your specific questions.
>
> **1. Faster rates:** The bounds in recent work by Mao et al. (2024a) are not wrong, but they are worst-case bounds that hold for any distribution. Specifically, these bounds rely on a fixed convex function $\Psi$ or concave function $\Gamma$, which is independent of both the distribution and the hypothesis.
>
> The introduction of the $\alpha$ and $\beta$ functions enable us to derive enhanced bounds with a non-constant factor $\gamma$, capturing properties of the distribution and the predictor $h$. Remarkably, under Tsybakov noise conditions, we can derive more favorable $H$-consistency bounds (see Theorem 6 and the comments below it) with better exponents. In the proof, we choose functions $\alpha$ and $\beta$ that depend on the input, the predictor $h$, and the best predictor $h^*$. For example, $\beta(h, x)$ measure the disagreement of $h$ and $h^*$ on $x$, modulo a small constant $\epsilon$.
>
> **2. Intuition behind functions $\alpha$ and $\beta$:** in prior work, the convex function $\Psi$ or concave function $\Gamma$ are fixed, they are independent of $h$ or any distribution-dependent quantity. Introducing the auxiliary functions $\alpha$ and $\beta$ helps us derive bounds that can capture distribution-dependent or predictor-dependent quantities, with a $\gamma$ factor.
>
> For example, in the enhanced $H$-consistency bounds with respect to the exponential loss in bipartite ranking (Theorem 12), the non-constant factor is the expected loss of a predictor based on an AdaBoost loss. This means that the rate of the bound becomes faster as the predictor gets closer to the best-in-class predictor. The best rate depends on the data distribution, as does the best-in-class expected loss.
>
> Similarly, in the enhanced $H$-consistency bounds with respect to the logistic loss in bipartite ranking (Theorem 13), the non-constant factor is the accuracy of the Bayes classifier. This means that the rate of the bound depends on the data distribution. In particular, in the deterministic case, the accuracy of the Bayes classifier is one.
>
> We cannot provide a general intuition for the choice of $\alpha$ and $\beta$, since a suitable choice seems to depend on the setting considered.
>
> **3. Approximation error:** as in standard discussions of estimation vs. approximation, the approximation error is the difference of the best-in-class expected loss and that of the best possible (Bayes) expected loss. We will add the definition: it is $\mathcal{E} _{\ell}^*(\mathcal{H}) - \mathcal{E} _{\ell}^*(\mathcal{H} _{\mathrm{all}})$.
>
> **4. Takeaways:** Our new fundamental tools enable the derivation of more favorable guarantees in various scenarios that
> 1) better leverage key distributional properties;
> 2) establish connections between existing algorithms; and
> 3) can lead to more favorable algorithms in other scenarios.
>
> For 1), an example is our more favorable $H$-consistency bounds under low-noise conditions for both binary and multi-class classification problems, with a linear rate when the Massart noise condition holds and an intermediate rate between linear and square-root for other values of $\alpha$ in the Tsybakov noise condition.
>
> For 2), an illustration is our enhanced $H$-consistency bounds in bipartite ranking theoretically explaining the empirical observation that AdaBoost exhibits a favorable ranking performance and a similar finding for the logistic loss.
>
> For 3), our new tools can be applied to comp-sum losses and the enhanced bounds can lead to better adversarial robustness algorithms, in a way similar to the derivation of better adversarially robust algorithms in [Mao et al., Cross-Entropy Loss Functions: Theoretical Analysis and Applications, 2023].

---

> > ### Comment · Reviewer_dSac · 2024-11-23
> >
> > Thank you to the authors for the thorough responses.
> >
> > There seems to be a lot of interesting ideas and connections being made here that I would like to understand (e.g. your response point 2; your takeaways). Still, I'm not sure I understand the paper much beyond what it said it accomplished. While I find the intuition in the rebuttal a bit vague, here is also probably not the place to expand on them.
> >
> > I recommend accepting this paper, with low confidence. If the paper is accepted, I hope the authors can add to the exposition.

---

### Meta-Review · Area_Chair_3xRq · 2024-12-11

**Recommendation:** Accept
**Confidence:** 3

**Metareview:**

The paper addresses the important and challenging topic of deriving fine-grained H-consistency bounds that are bounds the estimation error (relative to the best in H) rather than the total excess risk (relative to the best overall). The contributions include novel tools which allows to prove tighter bounds, with the previous H-consistency bounds being special case thereof. Various scenarios are considered, including standard multi-class classification, binary and multi-class classification under Tsybakov noise conditions, and bipartite ranking.

Overall, the paper has received a positive reception from reviewers and therefore I recommend the acceptance.

Please note that the reviewers suggested various improvements:
- Clarity of presentation and accessibility of the technical results for readers not deeply familiar with the previous works on this topic.
- Providing simple examples would be very welcome.
- Proof sketches for key theorems could be moved to the main text with an additional explanations to help readers understand the derivations.
- A more detailed comparison between the proposed bounds and earlier approaches (in particular, limitations of previous methods).
- Explain the intuition behind certain parameters and tools.

I recommend addressing these suggestions to improve the final version of the paper.

**Paper Award:**

No